# CAN LLMS RELIABLY EVALUATE THEMSELVES? A PROBABILISTIC VC FRAMEWORK

## ABSTRACT

As Large Language Models (LLMs) are increasingly deployed in autonomous reasoning tasks, the capacity to reliably evaluate their own outputs becomes paramount. We address this challenge by establishing a formal framework grounded in statistical learning theory. By operationalizing self-evaluation as a property of the hypothesis class induced by prompting strategies and stochastic decoding, we extend the classical Vapnik-Chervonenkis (VC) dimension to the probabilistic setting. We introduce two novel complexity measures: the Probabilistic VC (PVC) dimension, which quantifies the discriminative expressiveness of self-assessment, and the Calibration-aware PVC (C-PVC) dimension, which imposes a strict alignment constraint between confidence and correctness. In contrast to isolated calibration metrics, our unified framework provides integrated complexity measurements with provable generalization guarantees. A systematic evaluation of eleven 7–8B models across mathematical, factual, and commonsense domains highlights a fundamental trade-off: enhanced discriminative capacity systematically incurs a degradation in calibration quality. This structural tension suggests that current reasoning optimization paradigms do not implicitly resolve, and may exacerbate, miscalibration. Our framework offers the necessary diagnostic tools to quantify these risks, laying the groundwork for the development of trustworthy autonomous systems.

## 1 INTRODUCTION

A key requirement for advanced reasoning systems is not only the ability to solve complex tasks, but also the ability to evaluate the relative quality of their own reasoning processes. This self-evaluation capability, the ability to identify which of their generated solutions is superior, is essential for understanding model limitations and ensuring reliable autonomy. As large language models (LLMs) are increasingly applied to tasks requiring multi-step reasoning, a natural question arises: to what extent can these models discriminatively identify which of their generated solutions is of higher quality?

Recent progress in LLMs has led to significant improvements in reasoning performance across domains (Kojima et al., 2022; Wei et al., 2022; Chen et al., 2022), including mathematical problem solving (Li et al., 2024), pattern recognition (Nie et al., 2020; Zhang et al., 2019), and abstract reasoning (Chollet, 2019). These advances have been supported by preference-based fine-tuning techniques such as Direct Preference Optimization (DPO) (Rafailov et al., 2023) and Group Relative Policy Optimization (GRPO) (Shao et al., 2024; Guo et al., 2025; Yang et al., 2024a; Yu et al., 2025). However, existing evaluation protocols primarily focus on final answer accuracy, offering limited insight into a model's ability to assess its own reasoning quality.

This motivates our investigation into model self-evaluation capabilities, encompassing self-reflection, the internal mechanism through which a model assesses the relative quality of its own reasoning, among other aspects of metacognitive assessment. While recent studies (Shinn et al., 2023; Pan et al., 2023; Madaan et al., 2023; Toy et al., 2024; Wang & Zhao, 2023) have explored various self-evaluation heuristics, a principled framework for analyzing and comparing such capabilities remains lacking.

To address this gap, we propose a formal approach grounded in statistical learning theory that can systematically measure self-evaluation capacity. We extend the classical Vapnik–Chervonenkis (VC) dimension (Vapnik & Chervonenkis, 1971; Blumer et al., 1989) to the probabilistic setting where models must discriminate between their own solutions. We introduce the Probabilistic VC (PVC) dimension to quantify discriminative expressiveness, and the Calibration-aware PVC (C-PVC) dimension to enforce alignment between confidence and correctness. This unified framework addresses the probabilistic nature of modern LLMs while providing theoretical foundations for understanding the inherent trade-off between self-evaluation expressiveness and calibration quality.

## 1.1 OUR CONTRIBUTION

**Theoretical contributions.** We introduce the Probabilistic VC (PVC) dimension, extending classical capacity measures to quantify the expressiveness of probabilistic self-evaluation tasks where models must discriminate between solution qualities. Furthermore, we establish the Calibration-aware PVC (C-PVC) dimension, the first complexity measure that simultaneously captures discriminative power and calibration alignment. We prove that the PVC–C-PVC gap serves as a quantifiable metric for the calibration cost inherent in self-assessment, providing generalization bounds and sample complexity results that extend classical VC theory to the probabilistic setting.

**Methodological contributions.** We develop a practical framework for estimating these theoretical quantities through category-level aggregation, ensuring statistical rigor in empirical measurement. Our three-stage evaluation protocol, solution generation, self-selection, and ensemble validation, provides a systematic methodology for assessing self-evaluation capabilities. To address parameter dependence across confidence thresholds $\gamma$ and calibration tolerances $\tau$, we introduce the Volume Under Surface (VUS) aggregation. This yields parameter-free summary statistics, enabling robust cross-model comparisons and holistic analysis of the trade-off space.

**Empirical contributions.** We conduct the first systematic study of self-evaluation capabilities using complexity-theoretic measures across diverse reasoning domains. Our results identify a consistent trade-off between expressiveness and calibration across varied architectures, training paradigms (SFT, RLHF, distillation), and cognitive domains (mathematical, factual, commonsense reasoning). This suggests a fundamental characteristic of current probabilistic reasoning systems rather than an artifact of specific training procedures. These findings advance the theoretical understanding of self-reflection and provide diagnostic tools for principled model selection based on application-specific reliability requirements.

## 2 BACKGROUND

### 2.1 CLASSICAL VC DIMENSION

The Vapnik–Chervonenkis (VC) dimension (Vapnik & Chervonenkis, 1971; Blumer et al., 1989; Shalev-Shwartz & Ben-David, 2014) is a foundational measure of the capacity of a binary hypothesis class $\mathcal{H} \subseteq \{h : \mathcal{X} \to \{0,1\}\}$. A set $\{x_1, \ldots, x_n\} \subset \mathcal{X}$ is shattered by $\mathcal{H}$ if for every labeling $(y_1, \ldots, y_n) \in \{0,1\}^n$, there exists $h \in \mathcal{H}$ such that $h(x_i) = y_i$ for all $i$. The VC dimension, $\text{VC}(\mathcal{H})$, is the size of the largest such set, intuitively measuring how many points the hypothesis class can classify in all possible ways.

VC dimension is tightly linked to generalization. If $\text{VC}(\mathcal{H}) = d$, then for any $\delta \in (0, 1)$, with probability at least $1 - \delta$ over the random draw of a training sample of size $n$:

$$\mathbb{P}\left[\sup_{h \in \mathcal{H}} |R(h) - \hat{R}(h)| \leq C\sqrt{\frac{d + \log(1/\delta)}{n}}\right] \geq 1 - \delta,$$

where $C > 0$ is a universal constant, $R(h)$ is true risk (expected error on the population) and $\hat{R}(h)$ is empirical risk (average error on the sample). This implies a sample complexity of $\Omega\left(\frac{d + \log(1/\delta)}{\epsilon^2}\right)$ to achieve accuracy $\epsilon$ with confidence $1 - \delta$. However, classical VC theory is limited to binary-valued outputs, making it insufficient for analyzing LLMs that output confidence scores rather than hard decisions.

## 2.2 FAT-SHATTERING DIMENSION

The fat-shattering dimension (Bartlett & Long, 1995; Alon et al., 1997) generalizes VC theory from binary hypothesis classes $\mathcal{H}$ to real-valued function classes $\mathcal{F} \subseteq \{f : \mathcal{X} \to \mathbb{R}\}$, capturing how finely functions can distinguish inputs at a given scale $\alpha > 0$. A set $\{x_1, \ldots, x_n\}$ is $\alpha$-shattered by $\mathcal{F}$ if there exist thresholds $s_1, \ldots, s_n$ such that for all $\sigma \in \{\pm 1\}^n$, there exists $f \in \mathcal{F}$ satisfying:

$$\sigma_i(f(x_i) - s_i) \geq \alpha \quad \text{for all } i \in [n].$$

This condition means $f$ can place each output either $\alpha$ above or below its threshold specified by $\sigma_i$.

The fat-shattering dimension at scale $\alpha$, denoted $\text{fat}_\alpha(\mathcal{F})$, is the largest $n$ for which such a set exists. For binary functions, $\text{fat}_\alpha(\mathcal{F}) = \text{VC}(\mathcal{F})$ for all $\alpha \in (0, 1)$, and is zero for $\alpha \geq 1$. Fat-shattering serves as a scale-sensitive extension of VC dimension and provides the foundation for our probabilistic extension to LLMs (Alon et al., 1997; Bartlett et al., 1994; Bartlett & Long, 1995).

## 3 METHODS

We develop a theoretical framework for analyzing LLMs' self-evaluation capabilities through statistical learning theory. We extend classical VC theory to probabilistic predictors via two new complexity metrics: Probabilistic VC (PVC) dimension, which measures a model's ability to make confident classifications, and Calibration-aware PVC (C-PVC), which additionally requires alignment between confidence scores and actual success rates.

### 3.1 PROBABILISTIC VC DIMENSION AND ITS CALIBRATION-AWARE EXTENSION

In this paper, we use notation $\text{PVC}_\gamma(\mathcal{F})$ and $\text{C-PVC}_\gamma^\tau(\mathcal{F})$ when discussing theoretical definitions and general properties, and the abbreviated notation $\text{PVC}_\gamma$ and $\text{C-PVC}_\gamma^\tau$ when the function class is clear from context.

To analyze self-evaluation capabilities in language models, we extend the classical VC framework to accommodate probabilistic predictions. Rather than modeling hypotheses as deterministic binary functions, we consider predictors that output a distribution over possible labels.

**Definition 1** (Probabilistic VC Dimension). *Let $\mathcal{F}$ be a class of probabilistic predictors $f : \mathcal{X} \to \Delta(\{0, 1\})$, where $\Delta(\{0, 1\})$ denotes the set of probability distributions over the binary output space. Fix a correctness probability threshold $\gamma \in (0, 1]$. The probabilistic VC dimension $\text{PVC}_\gamma(\mathcal{F})$ is the largest integer $d$ such that there exists a set $\{x_1, \ldots, x_d\} \subseteq \mathcal{X}$ with the following property: for every labeling $y_1^*, \ldots, y_d^* \in \{0, 1\}$, there exists a predictor $f \in \mathcal{F}$ such that*

$$\mathbb{P}\left(f(x_i) = y_i^*\right) = \mathbb{E}\left[\mathbf{1}\{f(x_i) = y_i^*\}\right] \geq \gamma \quad \text{for all } i = 1, \ldots, d.$$

For example, if $\text{PVC}_{0.8}(\mathcal{F}) = 3$, then there exists a set of 3 inputs where the model can assign any binary labeling with at least 80% probability, but no set of 4 inputs can be so labeled. This relaxes classical shattering by requiring high-confidence probabilistic support rather than exact realizability. Importantly, this generalization aligns closely with the fat-shattering dimension (Alon et al., 1997; Bartlett et al., 1994; Bartlett & Long, 1995), and can be seen as a probabilistic analogue tailored for confidence-aware reasoning tasks.

While PVC quantifies the capacity for high-accuracy predictions, it does not ensure that the reported confidence scores reflect actual correctness probabilities. For high-stakes applications, calibration is crucial:

**Definition 2** ($\tau$-Calibration on Fixed Inputs). *Let $\{x_1, \ldots, x_d\} \subseteq \mathcal{X}$ be a set of fixed inputs and $y_1^*, \ldots, y_d^* \in \{0, 1\}$ be the corresponding true labels. A predictor-confidence pair $(f, \hat{p})$ is $\tau$-calibrated on these inputs if for every $i = 1, \ldots, d$:*

$$|\hat{p}(x_i) - \mathbb{E}\left[\mathbf{1}\{f(x_i) = y_i^*\}\right]| \leq \tau.$$

*Here, the expectation is taken over the internal randomness of the predictor $f$, while $\hat{p}(x_i)$ is treated as the reported confidence score for the input $x_i$.*

Unlike population-level metrics such as ECE (Guo et al., 2017), our definition requires point-wise calibration on fixed inputs. This aligns with the combinatorial nature of VC theory and imposes a strictly stronger constraint: it ensures calibration for each input individually, thereby preventing the cancellation of over- and under-confidence errors inherent in distributional averages.

This captures approximate calibration: a predictor is $\tau$-calibrated if its reported confidence deviates from the true correctness probability by at most $\tau$. We now integrate high-accuracy prediction and calibration:

**Definition 3** (Calibration-aware Probabilistic VC Dimension). *Let $\mathcal{F}$ be a class of probabilistic predictors $f : \mathcal{X} \to \Delta(\{0, 1\})$ and let $\hat{p} : \mathcal{X} \to [0, 1]$ denote a confidence scoring function associated with each $f \in \mathcal{F}$.*

*Fix accuracy threshold $\gamma \in (0, 1]$ and calibration error tolerance $\tau \in [0, 1)$. The calibration-aware probabilistic VC dimension $\text{C-PVC}_\gamma^\tau(\mathcal{F})$ is the largest integer $d$ such that there exists a **fixed set** $\{x_1, \ldots, x_d\} \subseteq \mathcal{X}$ with the following property:*

*For every labeling $(y_1^*, \ldots, y_d^*) \in \{0, 1\}^d$, there exists a pair $(f, \hat{p})$ such that:*

$$\mathbb{P}(f(x_i) = y_i^*) \geq \gamma, \quad and \quad |\hat{p}(x_i) - \mathbb{E}\left[\mathbf{1}\{f(x_i) = y_i^*\}\right]| \leq \tau \quad for \ all \ i = 1, \ldots, d.$$

## 3.2 LLMs as Function Classes

To apply our VC-theoretic framework to LLMs, we must interpret a single pretrained model not as a fixed function, but as a generator of a function class. We outline three complementary perspectives that justify this mapping:

**Bayesian view (parameter-level randomization).** Treat the parameter vector $W$ as a random variable $W \sim \mathcal{P}$, inducing the function class $\{f_W : W \in \text{supp}(\mathcal{P})\}$. This aligns with PAC-Bayesian analyses of randomized predictors (McAllester, 2003; Germain et al., 2016), the correspondence between infinitely wide neural networks and Gaussian processes (Lee et al., 2018; de G. Matthews et al., 2018), and non-vacuous deep-network generalization bounds via PAC-Bayes (Dziugaite & Roy, 2017).

**Prompt-based view (input-conditioning as a hypothesis family).** With fixed parameters $W$, varying a prompt $p$ from a constraint set $\Pi$ yields a hypothesis class:

$$\mathcal{F}_{\text{prompt}}(W) \ = \ \{ x \mapsto f_W([p; x]) \ : \ p \in \Pi \}.$$

This captures the intuition that prompting steers the model into specific reasoning modes (Li & Liang, 2021; Lester et al., 2021) and aligns with in-context learning interpretations (Xie et al., 2022). In our framework, distinct prompts correspond to distinct predictors $f \in \mathcal{F}$.

**Stochastic decoding view (output-level randomization).** Decoding randomness (e.g., temperature sampling, dropout) induces a distribution over outputs, formally mapping the model to a probabilistic predictor. This aligns with Gibbs predictors (McAllester, 2003; Germain et al., 2016) and Bayesian approximations (Gal & Ghahramani, 2015).

To rigorously operationalize the VC framework, we instantiate the hypothesis class $\mathcal{F}$ not as a single fixed model, but as a family of predictors induced by the interaction of prompts and decoding strategies. Formally, let $W$ denote the fixed parameters of the base model, $\Pi$ be the space of prompting strategies, and $\Omega$ be the space of stochastic decoding configurations (e.g., temperature seeds). We define the function class as $\mathcal{F} = \{f(\cdot; W, \pi, \omega) \mid \pi \in \Pi, \omega \in \Omega\}$. In our experimental evaluation, we empirically approximate this class by generating diverse reasoning paths ($\pi$) and utilizing temperature sampling ($\omega$) to realize the probabilistic nature of each predictor $f \in \mathcal{F}$.

## 3.3 Measuring PVC and C-PVC through Self-Evaluation

To empirically estimate PVC and C-PVC dimensions, we treat dataset categories (e.g., Algebra, Geometry) as the effective fixed inputs $\{x_1, \ldots, x_d\}$ required by Definition 3. To justify estimating probabilities via empirical averages, we assume that the questions within each category $C$ are sampled i.i.d. from a latent category-specific distribution $\mathcal{D}_C$. Due to the inherent variance in single-question evaluation, aggregating questions into categories provides a robust statistical estimate of the model's true performance probabilities. We employ a three-stage evaluation protocol (Figure 1): 1) The model

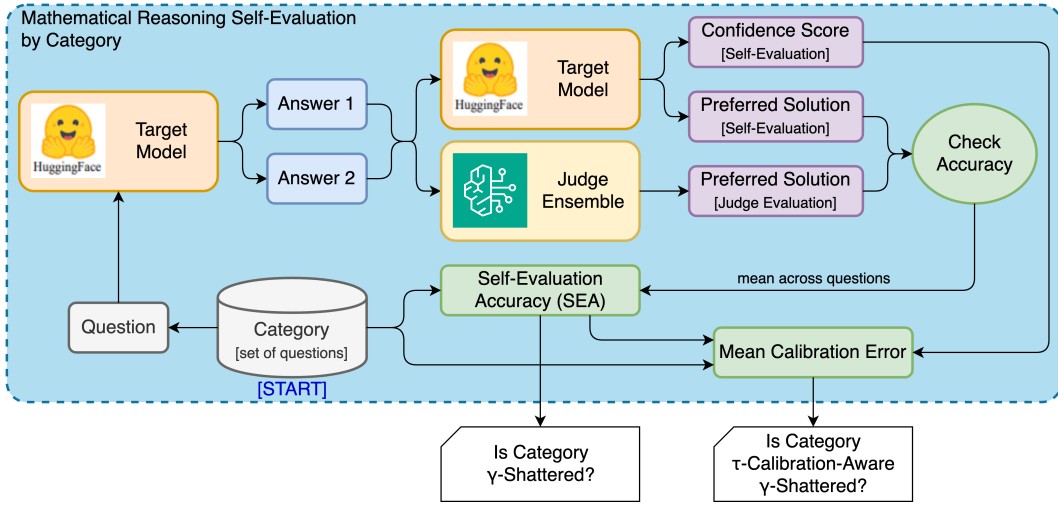

Figure 1: PVC/C-PVC Experiment Flow: Models generate two solutions, perform self-evaluation, and are compared against expert ensemble judgment.

generates two distinct solutions to each problem, 2) The model evaluates both solutions and reports its confidence, 3) External judge LLMs determine the objectively superior solution.

Our approach is based on two key metrics that approximate the theoretical conditions:

**Definition 4** (Self-Evaluation Accuracy). *Let $f \in \mathcal{F}$ be a probabilistic predictor, and $C$ a question category. For each question $q \in C$, let $\mathsf{Select}_f(q)$ be the model's choice (a random variable depending on $f$'s decoding) and $\mathsf{Judge}(q)$ be the ground truth. We estimate the correctness probability as:*

$$\mathrm{SEA}(f, C) = \mathbb{E}_{q \sim \mathcal{D}_C}\left[\mathbf{1}\left\{\mathsf{Select}_f(q) = \mathsf{Judge}(q)\right\}\right].$$

**Definition 5** (Calibration Error). *For a category $C$, let $\hat{p}(q)$ be the confidence score generated by $f$ for question $q$. We measure the alignment between the expected confidence and actual accuracy over the category:*

$$\mathrm{CalibError}(f, C) = \left|\mathbb{E}_{q \sim \mathcal{D}_C}[\hat{p}(q)] - \mathrm{SEA}(f, C)\right|.$$

Using these metrics, we approximate the capacity of the hypothesis class. A category $C$ is considered $\gamma$-shattered by predictor $f$ if $\mathrm{SEA}(f, C) \geq \gamma$. We denote the count of such shattered categories as the **Empirical PVC**, which serves as a practical lower bound estimate for the theoretical dimension of the induced function class $\mathcal{F}$. Similarly, for C-PVC, a category contributes to the estimate if it satisfies both the accuracy threshold $\mathrm{SEA}(f, C) \geq \gamma$ and the calibration constraint $\mathrm{CalibError}(f, C) \leq \tau$. This mapping allows us to quantitatively lower-bound the theoretical dimensions defined in Section 3.1.

### 3.4 Aggregation with Volume Under Surface (VUS)

A practical challenge in applying VC theory to LLMs is the dependence on threshold parameters $\gamma$ and $\tau$. Reviewing models at a single threshold can be misleading. To address this and provide a robust, parameter-free summary statistic, we adopt the Volume Under Surface (VUS) methodology. We extend the Area Under Curve (AUC) concept to three dimensions:

$$M\text{-VUS}(\mathcal{F}) = \iint_{\mathcal{G}} M(\gamma, \tau, \mathcal{F}) \, d\gamma \, d\tau \tag{1}$$

where $M \in \{\mathrm{PVC}, \mathrm{C\text{-}PVC}, \mathrm{SC}\}$ and the domain is $\mathcal{G} = (0, 1] \times [0, 1)$. This integration captures the model's aggregate capabilities across the entire trade-off space of accuracy and calibration requirements.

A high PVC-VUS indicates strong discriminative self-assessment across all confidence levels, while high C-PVC-VUS suggests the model maintains calibration even under strict tolerance constraints. For PVC-VUS, since the metric is independent of $\tau$, the integration effectively normalizes the score over the $\tau$ dimension, allowing direct comparison with C-PVC-VUS.

# 4 THEORETICAL RESULTS

This section presents the core theoretical foundations of our probabilistic VC (PVC) framework. We establish key relationships between different dimension variants, derive generalization guarantees, and demonstrate how our framework extends classical VC theory to the probabilistic setting of self-evaluating language models.

## 4.1 RELATIONSHIPS BETWEEN DIMENSION VARIANTS

A natural first question is how different variants of PVC dimension relate to each other and to classical VC dimension. The following proposition establishes these fundamental relationships:

**Proposition 1** (Relationships Between Dimension Variants). *For a class of predictors $\mathcal{F}$, assuming $\mathcal{F}$ includes deterministic predictors, the following relationships hold:*

*1. For any $\gamma_1, \gamma_2 \in (0, 1]$ where $\gamma_1 > \gamma_2$:*
$$\mathrm{VC}(\mathcal{F}) = \mathrm{PVC}_1(\mathcal{F}) \leq \mathrm{PVC}_{\gamma_1}(\mathcal{F}) \leq \mathrm{PVC}_{\gamma_2}(\mathcal{F}),$$

*2. For a fixed $\gamma \in (0, 1]$ and any $\tau_1, \tau_2 \in [0, 1)$ where $\tau_1 > \tau_2$:*
$$\mathrm{C\text{-}PVC}_\gamma^{\tau_2}(\mathcal{F}) \leq \mathrm{C\text{-}PVC}_\gamma^{\tau_1}(\mathcal{F}) \leq \mathrm{PVC}_\gamma(\mathcal{F}).$$

This proposition elucidates several important insights. First, classical VC dimension corresponds to PVC with perfect accuracy ($\gamma = 1$), establishing a clear connection to traditional learning theory where predictors are deterministic. Second, as we relax the accuracy threshold $\gamma$, the PVC dimension increases monotonically, reflecting the intuition that requiring lower probability of correctness allows the model to shatter more diverse patterns (i.e., greater expressivity).

In our analysis, we distinguish between two key parameters with fundamentally different roles. The parameter $\gamma$ specifies a minimum accuracy threshold that characterizes the expressive power of a hypothesis class (capacity). In contrast, $\delta$ (appearing in Throerem 1 below) represents a probabilistic upper bound on error in generalization analysis—the confidence level with which a learned predictor generalizes to unseen data.

A central finding of our work is that despite handling probabilistic outputs, the PVC framework yields generalization bounds that closely mirror those of classical VC theory. This connection becomes clear when we relate PVC dimension to the fat-shattering dimension, a well-established scale-sensitive extension of VC theory (Alon et al., 1997; Bartlett & Long, 1995; Colomboni et al., 2025). The detailed sample complexity bounds for the basic PVC dimension are provided in Appendix H, which includes the formal proof relating PVC to fat-shattering via a margin parameter $\alpha = \gamma - \frac{1}{2}$.

## 4.2 GENERALIZATION BOUNDS FOR CALIBRATION-AWARE PVC

While PVC quantifies the capacity for high-accuracy predictions, the calibration-aware extension provides stronger guarantees about both prediction accuracy and calibration quality:

**Theorem 1** (Generalization Bounds for C-PVC). *Let $\mathcal{F}$ be a class of predictor pairs $(f, \hat{p})$, where $f : \mathcal{X} \to \Delta(\{0, 1\})$ is a probabilistic predictor and $\hat{p} : \mathcal{X} \to [0, 1]$ is an associated confidence function.*

*For accuracy threshold $\gamma \in (0, 1]$ and calibration tolerance $\tau \in [0, 1)$, assume $\mathrm{C\text{-}PVC}_\gamma^\tau(\mathcal{F}) = d_{\gamma,\tau} < \infty$.*

*Then there exists a universal constant $K > 0$ such that for any $\epsilon, \delta \in (0, 1)$, if the sample size satisfies:*

$$m \geq \frac{K}{\epsilon^2} \left( d_{\gamma,\tau} + \log \frac{1}{\delta} \right), \tag{2}$$

*the following generalization guarantees hold uniformly across all $(f, \hat{p}) \in \mathcal{F}$ with probability at least $1 - \delta$ over the random draw of an i.i.d. sample $S = \{(x_i, y_i)\}_{i=1}^{m}$ from a distribution $\mathcal{D}$ over $\mathcal{X} \times \{0, 1\}$:*

$$\left| \frac{1}{m} \sum_{i=1}^{m} \mathbf{1}\{f(x_i) \neq y_i\} - \mathbb{P}_{(x,y)\sim\mathcal{D}}[f(x) \neq y] \right| \leq \epsilon \quad \textit{(prediction error gap)}$$

$$\left| \frac{1}{m} \sum_{i=1}^{m} (\hat{p}(x_i) - \mathbf{1}\{f(x_i) = y_i\}) - \mathbb{E}_{(x,y)\sim\mathcal{D}} [\hat{p}(x) - \mathbf{1}\{f(x) = y\}] \right| \leq \epsilon \quad \textit{(calibration error gap)}$$

**Interpretation of $\mathcal{D}$ vs. $\mathcal{D}_C$.** We model the data generation as a two-step process. First, a problem category $C$ (e.g., Algebra) is sampled from the global distribution $\mathcal{D}$. Second, specific questions $q$ are sampled from that category's internal distribution $\mathcal{D}_C$. Thus, our theoretical bounds guarantee that the model generalizes to unseen categories ($\mathcal{D}$), provided that we accurately measure performance within each category using samples from $\mathcal{D}_C$.

The proof relies on the fact that the C-PVC dimension bounds the fat-shattering dimension of the associated loss function classes. This enables the application of uniform convergence results for real-valued functions to both prediction accuracy and calibration error simultaneously.

## 5 EXPERIMENTS AND RESULTS

### 5.1 EXPERIMENTAL SETUP

To assess reasoning self-evaluation capabilities, we constructed a novel benchmark, Math-360, comprising 360 original math problems across 8 domains, stratified by difficulty and subcategory. This design avoids contamination from pretraining data and captures diverse reasoning patterns. To evaluate whether PVC and C-PVC generalize beyond mathematical reasoning, we extended our empirical analysis to two widely used benchmarks: TruthfulQA (Lin et al., 2022), which assesses factual correctness under adversarial phrasing, and CommonsenseQA (Talmor et al., 2019), which evaluates everyday commonsense reasoning. To align with our category-based estimation framework, we grouped the latter two datasets into 10 broad categories based on semantic topics and sampled 240 questions per benchmark. For further details on dataset construction, categorization, and problem distribution, refer to Appendix R.3 and Appendix R.4

We evaluate eleven 7–8B parameter models using standardized inference settings detailed in Appendix R.2. While models generated their own confidence scores, the ground-truth correctness of the selected solutions was determined using an ensemble of three larger LLMs (Claude 3.7 Sonnet (Anthropic, 2025), Amazon Nova Premier (Intelligence, 2024), and DeepSeek-R1 (Guo et al., 2025)) to mitigate individual biases. To validate the reliability of this automated judging protocol, we conducted a rigorous correlation analysis (see Appendix M), which demonstrates that the ensemble consensus significantly reduces variance compared to individual judges.

For Sample Complexity calculations, we standardize the universal constant $K = 1$, error tolerance $\epsilon = 0.1$, and confidence parameter $\delta = 0.05$. We emphasize that since the true theoretical constant $K$ is unknown, the resulting SC-VUS values should be interpreted as a Relative Complexity Index for comparing model classes, rather than as absolute prescriptions for required sample sizes.

### 5.2 PVC AND CALIBRATION PERFORMANCE

Table 1 presents the comparative analysis of model performance across three datasets. We report PVC-VUS to quantify expressive self-assessment capacity and C-PVC-VUS to incorporate calibration constraints.

**Expressivity-calibration trade-off.** Our results expose a consistent inverse relationship between expressive power and calibration quality, empirically validating Proposition 1. Models exhibiting strong discriminative capacity, such as s1.1-7B (PVC-VUS: 5.83) and Qwen2.5-7B-Instruct (5.61), systematically incur larger calibration penalties, resulting in wider PVC–C-PVC gaps (1.56 and 1.52, respectively). Conversely, models like JiuZhang3.0-7B prioritize calibration fidelity (lowest ECE: 0.209, smallest Gap: 0.95) at the expense of raw expressiveness (PVC-VUS: 4.79). This suggests

Table 1: Experimental results across three datasets (Math-360, TruthfulQA, CSQA). Values reported as: Math-360/TruthfulQA/CSQA. PVC-VUS Gap denotes the discrepancy between expressive capacity (PVC) and calibrated capacity (C-PVC). ECE is the Expected Calibration Error (lower is better), and AE (Actual Error) is the error rate of self-selection. Note: SC-VUS values are calculated with a standardized constant $K = 1$ and serve as a relative complexity index; absolute sample requirements depend on the specific distributional properties absorbed by $K$.

| Model | PVC-VUS ↑ | C-PVC-VUS ↑ | PVC-C-PVC-VUS Gap ↓ | SC-VUS ↑ | ECE ↓ | AE ↓ |
|---|---|---|---|---|---|---|
| Qwen2.5-7B (Pretrain) (Yang et al., 2024a) | 5.39 (4.78 / 5.66 / 5.73) | 3.76 (3.27 / 3.93 / 4.09) | 1.63 (1.51 / 1.73 / 1.64) | 676.5 (627.3 / 693.0 / 709.2) | 0.314 (0.320 / 0.318 / 0.303) | 0.422 (0.403 / 0.435 / 0.429) |
| Qwen2.5-7B-Instruct (SFT+RL) (Yang et al., 2024a) | 5.61 (4.65 / 6.13 / 6.04) | 4.09 (3.13 / 4.73 / 4.40) | 1.52 (1.52 / 1.40 / 1.64) | 708.3 (612.6 / 772.6 / 739.7) | 0.285 (0.337 / 0.247 / 0.290) | 0.401 (0.419 / 0.388 / 0.396) |
| Qwen2.5-Math-7B-Instruct (SFT+RL) (Yang et al., 2024b) | 4.70 (4.32 / 4.63 / 5.16) | 3.20 (3.10 / 2.95 / 3.56) | 1.50 (1.22 / 1.68 / 1.60) | 620.3 (610.1 / 595.2 / 655.6) | 0.335 (0.289 / 0.388 / 0.329) | 0.494 (0.461 / 0.538 / 0.483) |
| Llama-3.1-8B-Instruct (SFT+DPO) (Meta AI, 2024) | 4.55 (4.84 / 4.13 / 4.68) | 2.92 (3.51 / 2.44 / 2.80) | 1.63 (1.33 / 1.69 / 1.88) | 591.9 (651 / 544.4 / 580.2) | 0.385 (0.284 / 0.462 / 0.408) | 0.505 (0.394 / 0.588 / 0.533) |
| OpenThinker2-7B (SFT) (Open-Thoughts, 2023) | 5.37 (5.19 / 5.46 / 5.46) | 3.50 (3.68 / 3.36 / 3.47) | 1.87 (1.51 / 2.10 / 1.99) | 649.9 (667.5 / 635.6 / 646.6) | 0.357 (0.298 / 0.395 / 0.378) | 0.420 (0.351 / 0.456 / 0.454) |
| DeepSeek-R1-Distill-Qwen-7B (Distill) (Guo et al., 2025) | 5.10 (4.77 / 5.01 / 5.51) | 3.49 (3.24 / 3.40 / 3.83) | 1.61 (1.53 / 1.61 / 1.68) | 648.8 (624.3 / 639.5 / 683.0) | 0.339 (0.329 / 0.358 / 0.330) | 0.451 (0.404 / 0.500 / 0.450) |
| Bespoke-Stratos-7B (SFT) (Bespoke Labs, 2023) | 5.33 (4.44 / 5.67 / 5.89) | 3.54 (2.73 / 3.95 / 3.94) | 1.79 (1.71 / 1.72 / 1.95) | 654.0 (573.4 / 694.7 / 693.7) | 0.347 (0.388 / 0.315 / 0.338) | 0.430 (0.444 / 0.433 / 0.413) |
| JiuZhang3.0-7B (SFT) (Zhou et al., 2024) | 4.79 (4.49 / 4.55 / 5.34) | 3.84 (3.74 / 3.22 / 4.57) | 0.95 (0.75 / 1.33 / 0.77) | 684.4 (674.1 / 622.1 / 757.1) | 0.209 (0.169 / 0.309 / 0.149) | 0.484 (0.439 / 0.546 / 0.467) |
| Ministral-8B-Instruct-2410 (SFT+RL) (Mistral AI, 2024) | 4.88 (4.21 / 4.61 / 5.81) | 3.00 (2.50 / 2.65 / 3.85) | 1.88 (1.71 / 1.96 / 1.96) | 600.0 (549.6 / 565.0 / 685.4) | 0.402 (0.414 / 0.453 / 0.339) | 0.478 (0.472 / 0.542 / 0.421) |
| Open-Reasoner-Zero-7B (RL) (Hu et al., 2025) | 5.29 (4.52 / 5.47 / 5.88) | 3.50 (2.78 / 3.71 / 4.02) | 1.79 (1.74 / 1.76 / 1.86) | 650.6 (578.2 / 671.3 / 702.4) | 0.352 (0.388 / 0.341 / 0.327) | 0.434 (0.436 / 0.454 / 0.413) |
| s1.1-7B (SFT) (Muennighoff et al., 2025) | 5.83 (4.76 / 6.22 / 6.52) | 4.27 (3.22 / 4.71 / 4.88) | 1.56 (1.54 / 1.51 / 1.64) | 727.3 (622.5 / 771.4 / 788.0) | 0.285 (0.331 / 0.256 / 0.267) | 0.378 (0.407 / 0.379 / 0.348) |

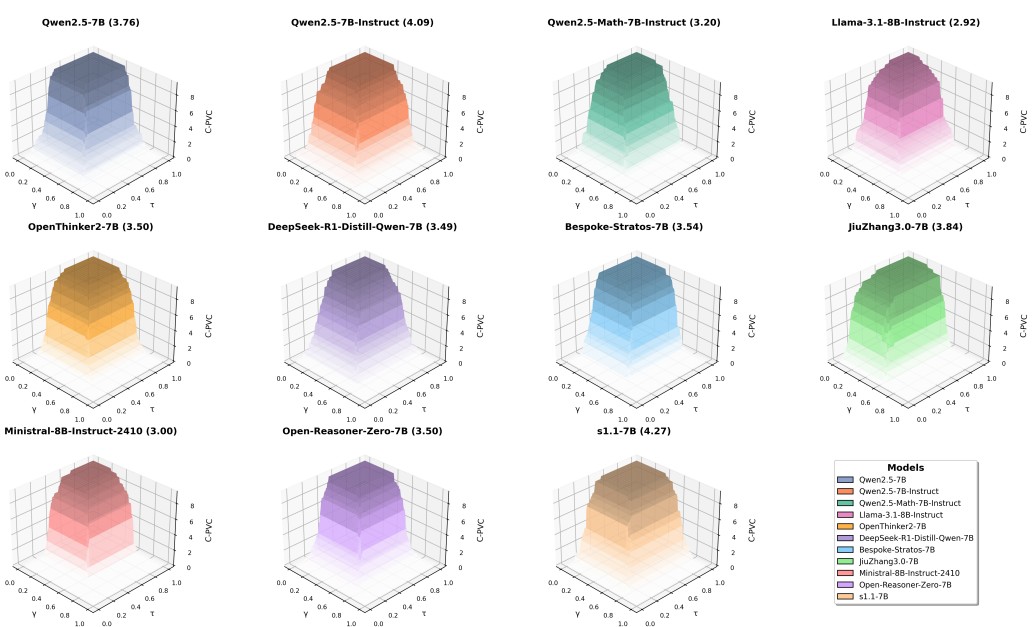

Figure 2: Calibration-aware Probabilistic VC (C-PVC) Dimension: 3D surface plots showing C-PVC dimension values for eleven 7-8B parameter language models. The x-axis ($\tau$) represents calibration error tolerance, y-axis ($\gamma$) shows accuracy threshold, and z-axis displays C-PVC values. Our framework quantifies self-evaluation capacity through these distinctive topographical signatures. Higher C-PVC-VUS (in parentheses) indicates better self-evaluation capabilities.

that current optimization techniques for reasoning capability do not implicitly solve, and may even exacerbate, the calibration problem.

**Sample complexity implications.** A critical insight from our framework is the quantification of generalization cost via SC-VUS. Notably, highly expressive models like s1.1-7B exhibit the highest SC-VUS (727.3), indicating that their superior capacity comes with a theoretical cost: they exhibit a higher relative structural complexity. This suggests that compared to models with lower SC-VUS,

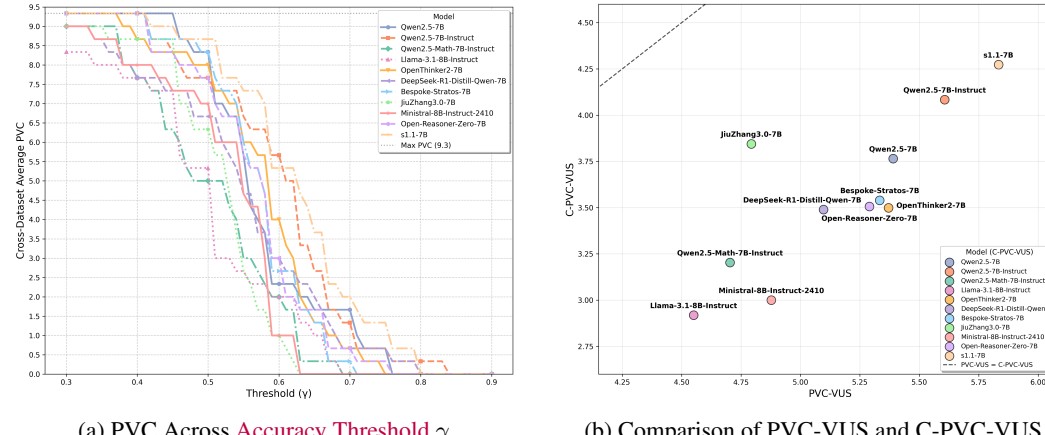

(a) PVC Across Accuracy Threshold $\gamma$      (b) Comparison of PVC-VUS and C-PVC-VUS

Figure 3: Model Self-Evaluation Capabilities and Calibration Performance: (a) PVC scores across accuracy thresholds ($\gamma$). Higher curves indicate better ability to maintain expressive reasoning as accuracy requirements increase. s1.1-7B and Qwen2.5-7B-Instruct show the strongest performance. (b) Comparison between PVC-VUS (expressiveness) and C-PVC-VUS (calibrated expressiveness). Points closer to the diagonal dashed line indicate better calibration. JiuZhang3.0-7B shows the smallest gap between these metrics, demonstrating superior calibration, while s1.1-7B achieves the highest overall expressiveness.

they theoretically necessitate proportionally more validation samples to distinguish genuine reasoning capability from stochastic noise or overfitting.

Instruction tuning generally enhances both PVC-VUS and C-PVC-VUS compared to base models, consistent with the expectation that explicit training improves discriminative capabilities. However, the persistence of the trade-off across all model families (Figure 3) indicates that standard instruction tuning alone may not sufficiently align confidence with correctness. Furthermore, our robustness analysis (Appendix K) suggests that this tension is structural rather than a superficial artifact of uncalibrated logits, as standard post-hoc methods like Temperature Scaling fail to effectively close the PVC–C-PVC gap. Figures 2 and 3b visualize this landscape, guiding practitioners to select models by prioritizing either discrimination (e.g., s1.1-7B) or trust (e.g., JiuZhang3.0-7B) based on specific reliability requirements.

## 5.3 CROSS-DOMAIN ANALYSIS

Our analysis highlights distinct patterns in self-evaluation capabilities. We observe clear domain-specific signatures, where models like Qwen2.5-7B-Instruct and Llama-3.1-8B-Instruct show specialized performance peaks in factual and mathematical domains, respectively, contrasting with the broad-spectrum consistency of s1.1-7B. Furthermore, the data suggests that calibration fidelity may operate as a transferable property distinct from raw discriminative power; for instance, JiuZhang3.0-7B maintains minimal PVC–C-PVC gaps across all datasets, indicating that the alignment mechanism is less sensitive to domain shifts than the reasoning process itself. Most significantly, the inverse correlation between expressivity and calibration is observed consistently across mathematical, factual, and commonsense reasoning. This empirical uniformity points to a fundamental characteristic of current probabilistic models, aligning with the theoretical generalization bounds of our PVC framework predicted by Theorem 1.

## 5.4 SELF-EVALUATION AS THE ATOMIC UNIT OF AUTONOMY

We analyze self-evaluation as the atomic unit of autonomous reasoning. In iterative agentic frameworks like Reflexion (Shinn et al., 2023), the global reliability of a system is functionally bounded by the calibration quality of its local decision nodes. Our protocol rigorously isolates this local decision step—discriminating superior solutions under uncertainty—without the confounding variables of planning or tool use. The observed trade-off between expressivity (PVC) and calibration (C-PVC)

identifies a systemic risk: while agents may possess the capacity to identify correct paths, the significant calibration deficits imply a high probability of silent failures, where agents mistakenly validate incorrect intermediate steps with high confidence. Consequently, C-PVC serves as a quantifiable lower bound for the reliability of autonomous decision-making loops.

## 6 RELATED WORKS

**LLM reasoning and optimization.** Large language models have demonstrated remarkable performance on complex reasoning tasks (Kojima et al., 2022; Wei et al., 2022; Chen et al., 2022), supported by benchmarks like MathVista (Li et al., 2024), Bongard-Logo (Nie et al., 2020), Raven (Zhang et al., 2019), and the ARC (Chollet, 2019). Techniques such as Direct Preference Optimization (Rafailov et al., 2023) and Group Relative Policy Optimization (Shao et al., 2024; Guo et al., 2025; Yang et al., 2024a; Yu et al., 2025) have further enhanced these capabilities. However, as Mondorf & Plank (2024) document, most frameworks prioritize final answer correctness over assessing reasoning quality itself.

**Calibration and reasoning reliability.** Calibration—the alignment between confidence scores and actual accuracy—has been extensively studied in deep learning (Guo et al., 2017; Ovadia et al., 2019) and LLMs (Kadavath et al., 2022). Recent research has actively leveraged these insights to enhance reasoning performance. For instance, Huang et al. (2025) proposed Adaptive Self-Consistency (ASC) to improve test-time scaling via self-calibration, while Zhou et al. (2025) introduced the Reasoning Probability Consistency (RPC) framework to theoretically bridge internal probabilities with self-consistency. However, a fundamental distinction exists: these works primarily focus on optimizing reasoning accuracy or explaining consistency mechanics. In contrast, our work focuses on metacognitive measurement. We do not aim to propose a new inference method to boost performance, but rather to establish a complexity-theoretic framework (PVC) that quantifies the intrinsic capacity of models to self-evaluate their own generated solutions.

**Theoretical foundations.** The Vapnik-Chervonenkis (VC) dimension (Vapnik & Chervonenkis, 1971; Blumer et al., 1989) provides the theoretical foundation for understanding model capacity and generalization (Shalev-Shwartz & Ben-David, 2014). Our work expands classical VC theory to probabilistic predictors by introducing calibration-aware extensions. This builds upon fat-shattering dimension theory (Alon et al., 1997; Bartlett & Long, 1995; Bartlett et al., 1994), which extends VC concepts to real-valued functions. By integrating these concepts, we establish direct connections between calibration quality and generalization guarantees, addressing a formalization gap in prior calibration studies.

For a comprehensive review, we refer readers to Appendix Q.

## 7 CONCLUSION

This paper establishes a principled framework for analyzing the self-assessment capabilities of large language models, introducing the Probabilistic VC (PVC) and Calibration-aware PVC (C-PVC) dimensions to quantify discriminative capacity and its alignment with reasoning validity. Our theoretical analysis extends classical learning theory to the probabilistic setting, providing rigorous sample complexity bounds and generalization guarantees.

Our empirical analysis exposes a fundamental tension in current reasoning systems: we identify a systematic trade-off across diverse architectures where enhanced discriminative power incurs a calibration cost, creating a significant PVC–C-PVC gap. This finding substantiates that scaling reasoning generation capabilities does not automatically yield reliable introspective reasoning.

Beyond theoretical analysis, our framework establishes a model selection criterion for system deployment, enabling practitioners to navigate the expressivity-trustworthiness trade-off based on application risks. We propose prioritizing models with minimal PVC–C-PVC gaps for safety-critical domains where silent failures are unacceptable, while leveraging models with maximal PVC dimensions for exploratory tasks where reasoning coverage is paramount. Ultimately, identifying C-PVC as a quantifiable safety gate for autonomous loops, we advocate for future research explicitly targeting the optimization of this trade-off to bridge the gap between reasoning performance and introspective reliability.

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

Table 2: Notation used throughout the paper.

| Symbol | Description |
|---|---|
| $\mathcal{X}$ | Input space (set of problem categories) |
| $\mathcal{F}$ | Class of probabilistic predictors $f : \mathcal{X} \to \Delta(\{0,1\})$ |
| $\Delta(\{0,1\})$ | Set of probability distributions over binary output space $\{0,1\}$ |
| $\gamma \in (0,1]$ | Accuracy threshold parameter |
| $\tau \in [0,1)$ | Calibration error tolerance parameter |
| $\mathrm{VC}(\mathcal{F})$ | Classical Vapnik-Chervonenkis dimension of function class $\mathcal{F}$ |
| $\mathrm{PVC}_\gamma(\mathcal{F})$ | Probabilistic VC dimension of $\mathcal{F}$ at accuracy threshold $\gamma$ |
| $\mathrm{C\text{-}PVC}_\gamma^\tau(\mathcal{F})$ | Calibration-aware PVC dimension with parameters $\gamma$ and $\tau$ |
| $\hat{p} : \mathcal{X} \to [0,1]$ | Confidence scoring function associated with predictor $f$ |
| $\mathrm{fat}_\alpha(\mathcal{F})$ | Fat-shattering dimension of function class $\mathcal{F}$ at scale $\alpha$ |
| $\alpha = \gamma - \frac{1}{2}$ | Margin parameter relating PVC to fat-shattering dimension |
| $\mathbb{P}(f(x) = y)$ | Probability that predictor $f$ outputs label $y$ on input $x$ |
| $\mathbb{E}[\cdot]$ | Expectation operator over the specified probability space |
| $\mathbf{1}\{\cdot\}$ | Indicator function: 1 if condition is true, 0 otherwise |
| $\mathrm{SEA}(f,C)$ | Self-Evaluation Accuracy of model $f$ on category $C$ |
| $\mathrm{Judge}(q)$ | Index of objectively superior solution for question $q$ |
| $\mathrm{Select}_f(q)$ | Model $f$'s selected solution index for question $q$ |
| $\mathrm{CalibError}(f,C)$ | Calibration error of model $f$ on category $C$ |
| $M\text{-}\mathrm{VUS}(\mathcal{F})$ | Volume Under Surface for metric $M \in \{\mathrm{PVC}, \mathrm{C\text{-}PVC}, \mathrm{SC}\}$ |
| $\mathrm{SC}(\gamma, \tau, \mathcal{F})$ | Sample complexity bound for function class $\mathcal{F}$ with parameters $\gamma, \tau$ |
| $\epsilon$ | Generalization error tolerance parameter |
| $\delta$ | Confidence parameter in generalization bounds |
| $K$ | Universal constant in sample complexity bounds |
| $\Pi$ | Set of prompts in prompt-based function class definition |
| $W$ | Parameter vector of a language model |
| $\mathcal{P}$ | Probability distribution over parameter space |
| $\mathcal{D}$ | Distribution over input space $\mathcal{X}$ (categories) |
| $\mathcal{D}_C$ | Distribution of questions within category $C$ |

## A    LIMITATIONS

Despite the theoretical and empirical contributions, this study has limitations inherent to its scope. First, our empirical estimation relies on category-level aggregation under the assumption that questions are sampled i.i.d. from a latent distribution ($\mathcal{D}_C$). While necessary for statistical robustness, this approach may mask fine-grained variances within broad categories. Second, our evaluation is confined to the 7–8B parameter regime. Although the expressivity-calibration trade-off is consistent within this scale, it remains an open question whether larger frontier models (e.g., >70B) might overcome this limitation through emergent capabilities. Third, our focus on the atomic unit of self-evaluation utilizes a static evaluation protocol. While this rigorously isolates the decision-making capability necessary for autonomy, it does not explicitly model the dynamic error propagation dynamics that occur in multi-turn agentic loops. Finally, our analysis of training methodologies is observational; establishing causal links between specific training objectives and PVC dimensions requires further controlled ablation studies.

## B    LLM USAGE DISCLOSURE

In the interest of transparency, we disclose that large language models (LLMs) were utilized as assistive tools during the preparation of this manuscript. Specifically, LLMs were employed to refine the clarity and readability of the text and to assist in writing code for data visualization. All theoretical formulations, experimental designs, data analyses, and scientific conclusions presented in this work are the original contributions of the authors. The authors have reviewed all content for accuracy and assume full responsibility for the final manuscript.

## C    PROOF OF PROPOSITION 1

*Proof.* We begin by proving the first statement: for any $\gamma_1, \gamma_2 \in (0, 1]$ such that $\gamma_1 > \gamma_2$, we have

$$\text{VC}(\mathcal{F}) = \text{PVC}_1(\mathcal{F}) \leq \text{PVC}_{\gamma_1}(\mathcal{F}) \leq \text{PVC}_{\gamma_2}(\mathcal{F}).$$

By definition, $\text{PVC}_\gamma(\mathcal{F})$ is the largest integer $d$ such that for every labeling $(y_1^*, \ldots, y_d^*)$, there exists $f \in \mathcal{F}$ satisfying

$$\mathbb{P}(f(x_i) = y_i^*) \geq \gamma \quad \text{for all } i = 1, \ldots, d.$$

When $\gamma = 1$, this condition requires $\mathbb{P}(f(x_i) = y_i^*) = 1$. Under the assumption that $\mathcal{F}$ contains deterministic predictors (or predictors capable of outputting probability 1), this coincides exactly with the classical VC shattering criterion, where each binary labeling must be realized exactly. Thus, we have

$$\text{PVC}_1(\mathcal{F}) = \text{VC}(\mathcal{F}).$$

Next, we consider the monotonicity of $\text{PVC}_\gamma$ with respect to $\gamma$. Suppose $\gamma_1 > \gamma_2$. Any set that is $\gamma_1$-shattered must also be $\gamma_2$-shattered, since the accuracy requirement $\gamma_2$ is weaker (i.e., easier to satisfy). Therefore,

$$\text{PVC}_{\gamma_1}(\mathcal{F}) \leq \text{PVC}_{\gamma_2}(\mathcal{F}),$$

and together with the previous identity, we obtain the desired chain of inequalities.

We now turn to the second statement: for a fixed $\gamma \in (0, 1]$ and any $\tau_1, \tau_2 \in [0, 1)$ where $\tau_1 > \tau_2$:

$$\text{C-PVC}_\gamma^{\tau_2}(\mathcal{F}) \leq \text{C-PVC}_\gamma^{\tau_1}(\mathcal{F}) \leq \text{PVC}_\gamma(\mathcal{F}).$$

First, we establish that $\text{C-PVC}_\gamma^{\tau_1}(\mathcal{F}) \leq \text{PVC}_\gamma(\mathcal{F})$. This follows directly from the definitions. The calibration-aware PVC dimension imposes two simultaneous conditions: for each $i = 1, \ldots, d$,

$$\mathbb{P}(f(x_i) = y_i^*) \geq \gamma \quad \text{and} \quad |\hat{p}_i - \mathbb{E}\left[\mathbf{1}\{f(x_i) = y_i^*\}\right]| \leq \tau_1,$$

where $\hat{p}_i$ is the confidence score assigned by $f$ to the input $x_i$. The first condition is identical to that of $\text{PVC}_\gamma$, but the second introduces an additional constraint that requires approximate calibration. As a result, any set that is $\text{C-PVC}_\gamma^{\tau_1}$-shattered is typically a subset of those that are $\text{PVC}_\gamma$-shattered (or at most equal in size). Thus, the dimension inequality holds.

Next, we show that $\text{C-PVC}_\gamma^{\tau_2}(\mathcal{F}) \leq \text{C-PVC}_\gamma^{\tau_1}(\mathcal{F})$ when $\tau_1 > \tau_2$. By definition, for a set to be $\text{C-PVC}_\gamma^\tau$-shattered, for every labeling, there must exist a predictor that achieves both the required accuracy $\gamma$ and calibration error at most $\tau$. Since $\tau_1 > \tau_2$, the calibration requirement for $\tau_2$ is stricter (tighter) than for $\tau_1$. Specifically, if a predictor satisfies

$$|\hat{p}_i - \mathbb{E}\left[\mathbf{1}\{f(x_i) = y_i^*\}\right]| \leq \tau_2,$$

then it automatically satisfies the condition for $\tau_1$ because $\tau_2 < \tau_1$. Therefore, any set that is $\text{C-PVC}_\gamma^{\tau_2}$-shattered must also be $\text{C-PVC}_\gamma^{\tau_1}$-shattered. This implies:

$$\text{C-PVC}_\gamma^{\tau_2}(\mathcal{F}) \leq \text{C-PVC}_\gamma^{\tau_1}(\mathcal{F}).$$

Combining these two inequalities completes the proof of the second statement.    $\square$

## D    SAMPLE COMPLEXITY BOUND VIA PROBABILISTIC VC DIMENSION

**Theorem 2** (Sample Complexity Bound via Probabilistic VC Dimension). *Let $\mathcal{F}$ be a class of probabilistic predictors over $\mathcal{X} \times \mathcal{Y}$ with probabilistic VC dimension $\text{PVC}_\gamma(\mathcal{F}) = d_\gamma < \infty$ for some accuracy threshold $\gamma \in (1/2, 1]$. Then there exist universal constants $K, c > 0$ such that the following holds. For any $\epsilon, \delta \in (0, 1)$, if the sample size satisfies*

$$m \geq \frac{K}{\epsilon^2}\left(d_\gamma + \log\frac{1}{\delta}\right),$$

*then*

$$\mathbb{P}_{S \sim \mathcal{D}^m}\left(\sup_{f \in \mathcal{F}}\left|\frac{1}{m}\sum_{i=1}^{m}\mathbf{1}\{f(x_i) \neq y_i\} - \mathbb{P}_{(x,y) \sim \mathcal{D}}[f(x) \neq y]\right| \leq \epsilon\right) \geq 1 - \delta.$$

*Proof.* Let each $f \in \mathcal{F}$ be a probabilistic predictor defined as a pair of measurable functions $(f_0, f_1) : \mathcal{X} \to [0, 1]$ such that $f_0(x) + f_1(x) = 1$ for all $x \in \mathcal{X}$. The prediction is interpreted as assigning label 1 when $f_1(x) > f_0(x)$ (which implies $f_1(x) > 1/2$), and label 0 otherwise. Thus, $f_1$ itself serves as a real-valued function mapping inputs to the probability assigned to label 1.

Define the function class $\mathcal{G} := \{f_1 : f = (f_0, f_1) \in \mathcal{F}\} \subseteq [0, 1]^{\mathcal{X}}$, and define the margin parameter $\alpha := \gamma - 1/2 > 0$ (valid since $\gamma > 1/2$). We now show that if $\mathrm{PVC}_\gamma(\mathcal{F}) = d_\gamma$, then $\mathcal{G}$ $\alpha$-fat-shatters a set of size at least $d_\gamma$ with fixed thresholds $s_i = 1/2$.

**Lemma 1** (PVC Implies Fat-Shattering with Fixed Threshold)**.** *Let $\mathcal{F}$, $\mathcal{G}$, and $\alpha$ be as above. Then*

$$\mathrm{PVC}_\gamma(\mathcal{F}) \leq \mathrm{fat}_\alpha(\mathcal{G}),$$

*where the fat-shattering condition is evaluated with fixed thresholds $s_i = 1/2$.*

*Proof.* Suppose $\{x_1, \ldots, x_d\}$ is $\gamma$-shattered in the PVC sense. Then for every labeling $(y_1^*, \ldots, y_d^*) \in \{0, 1\}^d$, there exists a probabilistic predictor $f = (f_0, f_1) \in \mathcal{F}$ such that for all $i = 1, \ldots, d$,

$$f_{y_i^*}(x_i) := \mathbb{P}(f(x_i) = y_i^*) \geq \gamma.$$

Consider $f_1 \in \mathcal{G}$ as the function assigning confidence to label 1. For each $i$, if $y_i^* = 1$, then $f_1(x_i) = f_{y_i^*}(x_i) \geq \gamma = 1/2 + \alpha = s_i + \alpha$. If $y_i^* = 0$, then $f_0(x_i) = f_{y_i^*}(x_i) \geq \gamma$ implies $f_1(x_i) = 1 - f_0(x_i) \leq 1 - \gamma = 1/2 - \alpha = s_i - \alpha$. Therefore, the set $\{x_1, \ldots, x_d\}$ is $\alpha$-fat-shattered by $\mathcal{G}$ at fixed thresholds $s_i = 1/2$. $\square$

We now apply the standard uniform convergence result for fat-shattering dimension:

**Lemma 2** (Uniform Convergence via Fat-Shattering (Colomboni et al., 2025))**.** *Let $\mathcal{G} \subseteq [0, 1]^{\mathcal{X}}$ and suppose $\mathrm{fat}_\alpha(\mathcal{G}) = d_\alpha < \infty$ for some $\alpha > 0$. Then there exist universal constants $K, c > 0$ such that for any $\epsilon, \delta \in (0, 1)$, if*

$$m \geq \frac{K}{\epsilon^2}\left(d_\alpha + \log\frac{1}{\delta}\right),$$

*then with probability at least $1 - \delta$ over i.i.d. samples $S \sim \mathcal{D}^m$,*

$$\sup_{g \in \mathcal{G}} \left|\frac{1}{m}\sum_{i=1}^{m} g(x_i) - \mathbb{E}[g(x)]\right| \leq \epsilon.$$

By Lemma 1, we have $\mathrm{PVC}_\gamma(\mathcal{F}) \leq \mathrm{fat}_\alpha(\mathcal{G})$ with $\alpha = \gamma - 1/2$. The generalization bound for the classification error of thresholded functions (binary predictions) is directly controlled by the fat-shattering dimension of the underlying real-valued function class $\mathcal{G}$ at margin $\alpha$ (see e.g., Anthony & Bartlett (2009), Theorem 14.1). Plugging the PVC dimension into this standard result yields the claimed sample complexity bound. $\square$

# E PROOF OF THEOREM 1

To rigorously establish the generalization bounds, we first explicate the relationship between the C-PVC dimension and the standard fat-shattering dimension, and specify the necessary structural assumptions on the loss function class.

## E.1 STRUCTURAL ASSUMPTIONS

We assume the following standard conditions for the function class $\mathcal{F}$:

1. **Boundedness:** For any pair $(f, \hat{p}) \in \mathcal{F}$, the confidence scoring function $\hat{p} : \mathcal{X} \to [0, 1]$ and the true conditional probability function $\eta(x) = \mathbb{P}(f(x) = y^*)$ are bounded in $[0, 1]$.

2. **Lipschitz Continuity of Loss:** The calibration error metric relies on the absolute difference loss $\ell(a, b) = |a - b|$, which is 1-Lipschitz continuous with respect to its arguments. This ensures that the covering number (and thus the complexity) of the loss class is controlled by the complexity of the underlying hypothesis class.

## E.2 REDUCTION TO FAT-SHATTERING DIMENSION

Recall that the fat-shattering dimension $\text{fat}_\alpha(\mathcal{G})$ at scale $\alpha$ measures the capacity of a real-valued function class $\mathcal{G}$ to deviate from a fixed threshold by at least $\alpha$.

**Lemma 3** (Fat-Shattering Bound via C-PVC). *Let $\mathcal{G}_{\text{conf}} := \{x \mapsto \hat{p}(x) \mid (f, \hat{p}) \in \mathcal{F}\}$ be the class of confidence scoring functions derived from $\mathcal{F}$. If the calibration-aware probabilistic VC dimension is finite, i.e., $C\text{-}PVC_\gamma^\tau(\mathcal{F}) = d_{\gamma,\tau} < \infty$, then the fat-shattering dimension of $\mathcal{G}_{\text{conf}}$ is bounded as:*

$$\text{fat}_\alpha(\mathcal{G}_{\text{conf}}) \leq d_{\gamma,\tau} \quad \text{for margin } \alpha = \gamma - \tau - 1/2, \tag{3}$$

*provided $\gamma - \tau > 1/2$.*

*Proof.* Fix any set of points $S = \{x_1, \dots, x_k\}$ that is $\alpha$-shattered by $\mathcal{G}_{\text{conf}}$ with fixed thresholds $s_i = 1/2$. This implies that for every binary labeling $b \in \{0, 1\}^k$, there exists a predictor $(f_b, \hat{p}_b) \in \mathcal{F}$ such that for all $i$:

- If $b_i = 1$, $\hat{p}_b(x_i) \geq 1/2 + \alpha = \gamma - \tau$.

- If $b_i = 0$, $\hat{p}_b(x_i) \leq 1/2 - \alpha = 1 - (\gamma - \tau)$.

Now consider the definition of C-PVC. For a set to be shattered in the C-PVC sense, we require $\mathbb{P}(f(x_i) = y_i^*) \geq \gamma$ and the calibration constraint $|\hat{p}(x_i) - \mathbb{P}(f(x_i) = y_i^*)| \leq \tau$. The calibration constraint implies $\hat{p}(x_i) \geq \mathbb{P}(f(x_i) = y_i^*) - \tau$. Combining this with the accuracy requirement ($\geq \gamma$), a valid C-PVC predictor must have confidence $\hat{p}(x_i) \geq \gamma - \tau$ for correct predictions. By setting $\alpha = \gamma - \tau - 1/2$, the condition for $\alpha$-fat-shattering becomes a necessary condition for C-PVC shattering. Thus, any set shattered by C-PVC logic can be $\alpha$-fat-shattered by the confidence functions. Therefore, $fat_\alpha(\mathcal{G}_{\text{conf}})$ is bounded by $C\text{-}PVC_\gamma^\tau(\mathcal{F})$. $\square$

## E.3 PROOF OF GENERALIZATION BOUNDS

*Proof of Theorem 1.* The theorem asserts uniform convergence for both prediction error and calibration error.

**1. Prediction Error Bound:** The prediction accuracy is governed by the binary classification capacity. From Proposition 1, we verify that $C\text{-}PVC_\gamma^\tau(\mathcal{F}) \leq PVC_\gamma(\mathcal{F})$. As established in classical theory (e.g., Anthony & Bartlett, 2009), the finite VC-style dimension of the thresholded classifiers implies uniform convergence of the empirical risk to the true risk.

**2. Calibration Error Bound:** We aim to bound the generalization gap for the calibration term. By applying the triangle inequality, the uniform deviation can be decomposed into two distinct estimation errors:

$$\sup_{(f,\hat{p}) \in \mathcal{F}} \left| \frac{1}{m} \sum_{i=1}^{m} (\hat{p}(x_i) - \mathbf{1}\{f(x_i) = y_i\}) - \mathbb{E}[\hat{p}(x) - \mathbf{1}\{f(x) = y\}] \right|$$

$$\leq \underbrace{\sup_{\hat{p} \in \mathcal{G}_{\text{conf}}} \left| \frac{1}{m} \sum_{i=1}^{m} \hat{p}(x_i) - \mathbb{E}[\hat{p}(x)] \right|}_{\text{(A) Confidence estimation error}} + \underbrace{\sup_{f \in \mathcal{F}} \left| \frac{1}{m} \sum_{i=1}^{m} \mathbf{1}\{f(x_i) = y_i\} - \mathbb{P}[f(x) = y] \right|}_{\text{(B) Accuracy estimation error}}$$

We bound each term separately using the complexity measures:

- **Term (A):** The complexity of the confidence function class $\mathcal{G}_{\text{conf}}$ is controlled by its fat-shattering dimension $\text{fat}_\alpha(\mathcal{G}_{\text{conf}})$. As shown in Lemma 3, this is bounded by $d_{\gamma,\tau}$. Thus, standard uniform convergence for real-valued functions applies.

- **Term (B):** The complexity of the correctness indicator is controlled by the standard PVC dimension. Since both complexity terms are finite, standard VC bounds apply.

Since both terms (A) and (B) are bounded by terms of the order $O(\sqrt{d/m})$, their sum is also bounded by the same order. Specifically, with probability at least $1 - \delta$, the total calibration generalization gap satisfies:

$$\text{Total Error} \leq K\sqrt{\frac{d_{\gamma,\tau}\log^2(m) + \log(1/\delta)}{m}}$$

where $K$ is a universal constant that absorbs the constants from both terms. By ensuring this total deviation is at most $\epsilon$, we solve for $m$ to derive the sample complexity bound stated in Theorem 1:

$$m \geq \frac{K}{\epsilon^2}\left(d_{\gamma,\tau} + \log\frac{1}{\delta}\right)$$

Note: The polylogarithmic factor $\log^2(m)$ is typically absorbed into the constant $K$ or treated as a lower-order term in standard sample complexity statements. □

## F  IMPLICATIONS FOR SELF-EVALUATING MODELS

Our theoretical results in Section 4 provide the first rigorous complexity measures for self-evaluating language models. By deriving VC-style bounds for probabilistic predictors, we bridge the gap between classical learning theory and modern confidence-aware reasoning.

The practical utility of our framework lies in the precise differentiation between discriminative capacity (PVC) and trustworthiness (C-PVC). A high PVC dimension indicates that a model possesses the raw latent capability to distinguish between superior and inferior solutions. However, without high C-PVC, this capability remains latent and potentially dangerous; a model may correctly identify solutions but assign overconfident probabilities to incorrect ones, leading to "hallucinated certainty." Conversely, ensuring high C-PVC aligns confidence with correctness, satisfying a prerequisite for trustworthy autonomous reasoning (Jiang et al., 2022; Kirsch & Gal, 2022; Shabat et al., 2020).

Consequently, the PVC–C-PVC gap emerges as a critical diagnostic metric, quantifying the "calibration cost" required to convert raw reasoning power into reliable self-assessment. This theoretical decomposition allows practitioners to diagnose whether a system failure stems from a lack of knowledge (low PVC) or a failure of introspection (large gap), guiding targeted interventions such as distinct training for reasoning versus calibration.

The trade-off between expressivity and calibration quality emerges naturally from this theoretical analysis and, as demonstrated in Section 5, manifests consistently across different model architectures.

## G  EXPECTED CALIBRATION ERROR AND CALIBRATION METRICS

In Section 3.3, we defined the category-level calibration error. To maintain consistency with our probabilistic framework, we denote the expectation over questions as sampling from the category-specific distribution $\mathcal{D}_C$:

$$\text{CalibError}(f, C) = |\mathbb{E}_{q\sim\mathcal{D}_C}[\hat{p}(q)] - \text{SEA}(f, C)| \tag{4}$$

where $\hat{p}(q)$ is the confidence score and $\text{SEA}(f, C)$ is the expected accuracy. This metric serves as the atomic unit for the calibration constraint in C-PVC.

Building on this, we define the Category-based Expected Calibration Error (ECE) as the weighted average of these local errors:

$$\text{ECE} = \sum_{C \in \mathcal{C}} \frac{|C|}{N} \text{CalibError}(f, C) = \sum_{C \in \mathcal{C}} \frac{|C|}{N} \left| \mathbb{E}_{q \sim \mathcal{D}_C}[\hat{p}(q)] - \text{SEA}(f, C) \right| \tag{5}$$

Additionally, the Actual Error (AE) measures the global failure rate of self-evaluation:

$$\text{AE} = \sum_{C \in \mathcal{C}} \frac{|C|}{N} (1 - \text{SEA}(f, C)) \tag{6}$$

where $\mathcal{C}$ is the set of categories, $|C|$ is the sample size per category, and $N$ is the total number of samples.

**Remark 1** (Justification for Category-based ECE). *Standard ECE typically groups predictions by confidence bins (e.g., $[0.1, 0.2]$). However, our Category-based ECE groups predictions by semantic domains (categories). This aligns with our theoretical premise that "shattering" occurs at the category level. By averaging calibration errors over categories rather than confidence bins, we directly link the global ECE metric to the local constraints used in C-PVC dimensions.*

### G.1 RELATIONSHIPS BETWEEN EVALUATION METRICS

Our empirical results in Table 1 reveal structural relationships between our complexity measures and traditional metrics.

**PVC-C-PVC-VUS Gap vs. ECE (The Calibration Cost).** We observe a strong positive correlation between the PVC–C-PVC Gap and ECE. This relationship is structurally inherent: the PVC-C-PVC-VUS Gap quantifies the loss of recognized capacity when enforcing strict calibration, while ECE measures the average magnitude of miscalibration. Mathematically, a large ECE implies that for many categories, the discrepancy $|\text{Conf} - \text{Acc}|$ is significant. This makes it difficult to satisfy the C-PVC constraint (CalibError $\leq \tau$) unless the tolerance $\tau$ is set to a very high value, thereby reducing the integrated volume (C-PVC-VUS) and increasing the Gap. Consequently, the Gap serves as a complexity-theoretic proxy for the empirical calibration error.

**SC-VUS vs. AE (Capacity vs. Performance).** The relationship between Sample Complexity (SC-VUS) and Actual Error (AE) reflects the classic trade-off between model capacity and empirical performance. Here, SC-VUS aggregates the theoretical VC dimensions to serve as a proxy for model capacity, while AE measures the realized empirical error. High SC-VUS indicates a high-capacity model capable of shattering many categories (high expressiveness). Theoretically, higher capacity allows for lower empirical error (Low AE), provided the model is well-trained. This dynamic explains why models like s1.1-7B exhibit both the highest SC-VUS (727.3) and the lowest AE (0.378). However, capacity is a double-edged sword; without the calibration constraints captured by the Gap, high capacity can lead to overconfident errors rather than reliable reasoning.

### G.2 RELATIONSHIP BETWEEN ECE AND C-PVC

While both metrics assess calibration, they operate on fundamentally different principles. C-PVC functions as a hard constraint, counting only those categories that strictly satisfy the calibration tolerance $\tau$, thereby measuring the breadth of reliable domains. In contrast, ECE acts as a soft average, quantifying the mean deviation across all domains. Consequently, low ECE is a necessary but not sufficient condition for high C-PVC.

To illustrate why this distinction matters, consider a model evaluated on two equally weighted categories: Factual Knowledge (50% of data) and Reasoning (50% of data). Suppose the model

is perfectly calibrated on Factual Knowledge (0 error) but moderately miscalibrated on Reasoning (0.14 error) with a tolerance of $\tau = 0.1$. In this scenario, the weighted average ECE would be $0.5 \times 0 + 0.5 \times 0.14 = 0.07$. An ECE of 0.07 is typically considered excellent in LLM benchmarks, creating an illusion of high reliability. However, under the C-PVC framework, the Reasoning category violates the tolerance (CalibError $0.14 > \tau$) and fails to shatter. This failure effectively halves the model's C-PVC dimension despite the low ECE. This example demonstrates that ECE can mask significant reliability failures in specific domains by averaging them with high-performing ones, whereas C-PVC demands uniform reliability across all shattered domains. The superior performance of JiuZhang3.0-7B in both metrics confirms that it achieves robust calibration without relying on such averaging effects.

## H INTERPRETATION OF SAMPLE COMPLEXITY

We provide a detailed interpretation of the sample complexity bounds derived in our PVC framework, linking theoretical quantities to the practical constraints of training and evaluating self-reflective LLMs.

### H.1 THEORETICAL INTERPRETATION OF BOUNDS

As established in Theorem 2 and Theorem 1, the sample complexity is governed by the interplay between the accuracy threshold $\gamma$ and the calibration tolerance $\tau$. The parameter $\gamma$ dictates the baseline expressiveness; typically, lower values of $\gamma$ yield higher dimensions ($d_\gamma$) as models can shatter more categories when the requirement for correctness probability is relaxed. This captures the fundamental intuition that guaranteeing high-confidence correctness is significantly cheaper in terms of sample complexity than validating weak reasoning capabilities.

Conversely, in the calibration-aware setting, the tolerance $\tau$ imposes a structural constraint on the effective hypothesis class. As formally expressed in the bound:

$$m \geq \frac{K}{\epsilon^2} \left( d_{\gamma,\tau} + \log \frac{1}{\delta} \right), \tag{7}$$

enforcing strict calibration (low $\tau$) acts as a strong regularizer. This constraint reduces the effective dimension relative to the uncalibrated baseline:

$$d_{\gamma,\tau} \leq d_\gamma.$$

By filtering out hypotheses that fail to align confidence with correctness, the observed PVC–C-PVC gap explicitly quantifies the complexity penalty ($d_\gamma - d_{\gamma,\tau}$) incurred to satisfy this reliability constraint.

### H.2 PRACTICAL IMPLICATIONS FOR LLM EVALUATION

These theoretical mechanics position the SC-VUS metric as a quantifiable proxy for the verification cost of self-assessment capabilities. Models exhibiting high SC-VUS, such as s1.1-7B, demonstrate substantial potential expressiveness but incur a significant generalization overhead. This necessitates extensive validation samples to distinguish genuine reasoning capability from stochastic noise or overfitting.

In contrast, models like JiuZhang3.0-7B, which maintain a minimal PVC–C-PVC gap, exemplify calibration efficiency. Their expressiveness is structurally regularized by the calibration constraint, rendering them more sample-efficient in establishing rigorous performance bounds. Furthermore, our cross-domain analysis indicates that this calibration quality is a transferable property. This supports a hierarchical training strategy: optimizing for calibration in data-rich domains (e.g., math) to facilitate reliability transfer to data-scarce domains. Theoretical bounds suggest that such well-calibrated models—characterized by lower effective complexity for a fixed error tolerance $\epsilon$—are better conditioned to generalize robustly under distribution shifts ($\mathcal{D} \rightarrow \mathcal{D}'$).

# I  CATEGORY-LEVEL MEASUREMENT FOR PVC AND C-PVC

In our analysis, we compute $\text{PVC}_\gamma(\mathcal{F})$ and $\text{C-PVC}_\gamma^\tau(\mathcal{F})$ at the level of problem categories (e.g., Algebra, Geometry) rather than individual questions. This design choice is motivated by theoretical and practical considerations, ensuring that our empirical estimates align with the probabilistic nature of the defined dimensions.

**Statistical tractability and effective hypothesis space.** Estimating VC-style capacity at the question level is computationally intractable due to the exponential growth of binary labelings ($2^n$). By defining the input space $\mathcal{X}$ as the set of semantic categories, we reduce the effective hypothesis space to a manageable finite set while preserving semantic diversity. This aggregation treats each category as a "macro-input," enabling efficient computation of shattering without sacrificing the ability to discriminate between models with different generalization behaviors.

**Alignment with model organization.** LLMs tend to acquire reasoning skills at the domain level before mastering specific problem templates. Measuring shattering capacity over category partitions aligns with this inductive bias. Each category acts as a higher-order instance representing a distribution $\mathcal{D}_C$, requiring the model to succeed across a range of related problems rather than overfitting to a single input.

**Stability in calibration estimation.** Individual prediction confidences are inherently noisy. As defined in Appendix G, we estimate the probability of correctness for a "category-input" $x_i$ via the expectation over the category-specific distribution $\mathcal{D}_{x_i}$. This averaging yields a robust estimate of the model's true confidence alignment:

$$\left| \mathbb{E}_{q \sim \mathcal{D}_{x_i}}[\hat{p}(q)] - \text{SEA}(f, x_i) \right| \leq \tau.$$

Evaluating this quantity at the category level mitigates the variance of single-question estimates, providing a stable signal for the C-PVC constraint.

**Theoretical consistency with $\gamma$-shattering.** Importantly, this formulation remains consistent with Definition 1. When we map an "input" $x_i$ to a "category" $C$, the probabilistic condition $\mathbb{P}(f(x_i) = y_i^*) \geq \gamma$ translates to requiring that the expected accuracy over the category meets the threshold:

$$\text{SEA}(f, x_i) = \mathbb{E}_{q \sim \mathcal{D}_{x_i}} \left[ \mathbf{1}\{\text{Select}_f(q) = \text{Judge}(q)\} \right] \geq \gamma.$$

Instead of demanding uniform correctness on every atomic question (which is overly restrictive and noisy), this approach evaluates whether the model has reliably "solved" the domain $x_i$ with probability at least $\gamma$. This yields a valid and robust estimate of the model's expressive capacity over the space of reasoning domains.

# J  ON THE FINITE CAPACITY GUARANTEES FOR LLMs

According to Anthony and Bartlett (Anthony & Bartlett, 2009, See Theorem 14.18 and Theorem 14.19), probabilistic VC capacity remains finite for model classes whose prediction functions are realized by bounded-weight, Lipschitz neural networks. In particular, the finiteness of both $\text{PVC}_\gamma$ and $\text{C-PVC}_\gamma^\tau$ follows from Lemma 1, specifically the existence of a finite fat-shattering dimension:

$$\text{C-PVC}_\gamma^\tau(\mathcal{F}) \leq \text{PVC}_\gamma(\mathcal{F}) \leq \text{fat}_\alpha(\mathcal{G}) < +\infty,$$

where $\alpha = \gamma - \frac{1}{2}$ and $\mathcal{G}$ is the associated class of real-valued confidence functions. This implication is powerful in its generality but depends critically on structural conditions—bounded weight norms, controlled depth, and uniformly Lipschitz activations—that are only partially satisfied in practice.

**The gap between theoretical assumptions and practical architectures.** While the bounded-operator and Lipschitz assumptions hold for simplified multi-layer perceptrons (MLPs), their applicability to realistic large language models (LLMs), such as those built on the Transformer architecture, remains unclear. Modern LLMs incorporate residual pathways, softmax-based attention, layer

normalization, and positional encodings, each of which may violate or obscure the assumptions required for a clean fat-shattering analysis. Although recent efforts attempt to bound the Lipschitz constants of certain attention modules and residual networks, no complete result exists that establishes fat-shattering finiteness for the full Transformer class. That is, the following capacity inequality,

$$\text{fat}_\alpha(\mathcal{G}_{\text{Transformer}}) < \infty,$$

remains a conjecture rather than a provable property under current theoretical tools. This gap complicates the ability to rigorously assert finite probabilistic capacity for the very models used in practice, especially when considering their introspective reliability and calibration behavior.

**Gaussian processes and infinite-dimensional predictors.** A similar gap arises for Gaussian Process (GP) models, where the predictive function is drawn from a stochastic process defined by a positive-definite kernel. To the best of our knowledge, while each realization of a GP resides in a Reproducing Kernel Hilbert Space (RKHS), and RKHS complexity measures such as covering numbers or Rademacher complexity are known to relate to generalization (Bartlett & Mendelson, 2002), the fat-shattering dimension of the GP function class is not known in general. In particular, there is no general result guaranteeing that

$$\text{fat}_\alpha(\mathcal{G}_{\text{GP}}) < \infty$$

for any fixed margin $\alpha > 0$, unless additional smoothness or norm constraints are imposed on the sample paths or kernels. This limits the applicability of PVC-style generalization theory to GP-based uncertainty quantification.

**Role of category-level shattering.** Measuring $\text{PVC}_\gamma$ at the category (rather than the individual-question) level restricts the effective domain of each predictor to the finite index set $\{\mathcal{D}_1, \ldots, \mathcal{D}_M\}$, where $M$ denotes the number of categories in the benchmark. For any predictor $f = (f_0, f_1) \in \mathcal{F}$, define the aggregated confidence

$$\mathbb{P}\big(f(x) = y^*(x)\big), \qquad c = 1, \ldots, M.$$

Consequently, the probabilistic VC dimension satisfies the immediate bound

$$\text{PVC}_\gamma(\mathcal{F}) \leq M,$$

and, by definition, the same upper bound holds for $\text{C-PVC}_\gamma^\tau(\mathcal{F})$. The finiteness of PVC thus follows without relying on global weight-norm or Lipschitz constraints.

**Outlook and open directions.** Notwithstanding the function-class perspective of Section 3.2, the fat-shattering framework—while principled for bounded, Lipschitz neural networks—has yet to be rigorously extended to modern LLM and kernel-based architectures. At present, finiteness of $\text{PVC}_\gamma$ and $\text{C-PVC}_\gamma^\tau$ is provable only under relatively restrictive assumptions, such as

$$\|W^{(\ell)}\|_1 \leq B_\ell \quad \text{and} \quad \phi \text{ is } \rho\text{-Lipschitz}$$

for every layer $\ell$. In contrast, models used in practice may exceed these bounds or include non-Lipschitz operations. Developing theoretical tools that bound the probabilistic capacity of richer, structured architectures—particularly Transformers and Gaussian Processes—therefore remains an important direction for future work. Such advances would narrow the gap between introspection theory and the empirical performance of high-capacity predictors in contemporary systems.

## K    DISCUSSION ON ROBUSTNESS TO POST-HOC CALIBRATION

One might ask whether the observed trade-off is merely an artifact of uncalibrated logits that could be resolved by Temperature Scaling (Guo et al., 2017). However, we argue that the expressivity-calibration tension captured by our framework is structural. Temperature Scaling applies a global monotonic transformation to confidence scores, which can improve global ECE but cannot simultaneously rectify conflicting misalignments across different categories (e.g., being overconfident in Algebra but underconfident in Geometry). Since C-PVC requires shattering (validity) across all

Table 3: Experimental results on the MATH-500 dataset. Consistent with Table 1, PVC-C-PVC-VUS Gap quantifies the calibration cost, and SC-VUS measures theoretical sample complexity. Note: As in Table 1, SC-VUS is a relative index computed with $K = 1$; actual sample bounds scale with the unknown constant $K$.

| Model | PVC-VUS ↑ | C-PVC-VUS ↑ | PVC-C-PVC-VUS Gap ↓ | SC-VUS ↑ | ECE ↓ | AE ↓ |
|---|---|---|---|---|---|---|
| Qwen2.5-7B (Pretrain) (Yang et al., 2024a) | 3.93 | 2.56 | 1.37 | 555.6 | 0.356 | 0.439 |
| Qwen2.5-7B-Instruct (SFT+RL) (Yang et al., 2024a) | 3.94 | 2.54 | 1.39 | 554.3 | 0.358 | 0.438 |
| Qwen2.5-Math-7B-Instruct (SFT+RL) (Yang et al., 2024b) | 3.68 | 2.57 | 1.11 | 556.8 | 0.304 | 0.476 |
| Llama-3.1-8B-Instruct (SFT+DPO) (Meta AI, 2024) | 3.97 | 2.67 | 1.30 | 566.7 | 0.331 | 0.435 |
| OpenThinker2-7B (SFT) (Open-Thoughts, 2023) | 4.36 | 3.09 | 1.26 | 609.4 | 0.294 | 0.378 |
| DeepSeek-R1-Distill-Qwen-7B (Distill) (Guo et al., 2025) | 3.95 | 2.52 | 1.43 | 551.6 | 0.369 | 0.437 |
| Bespoke-Stratos-7B (SFT) (Bespoke Labs, 2023) | 4.19 | 2.88 | 1.30 | 588.4 | 0.330 | 0.403 |
| JiuZhang3.0-7B (SFT) (Zhou et al., 2024) | 3.74 | **3.11** | **0.63** | **610.9** | **0.169** | 0.466 |
| Ministral-8B-Instruct-2410 (SFT+RL) (Mistral AI, 2024) | 3.97 | 2.50 | 1.46 | 550.3 | 0.377 | 0.434 |
| Open-Reasoner-Zero-7B (RL) (Hu et al., 2025) | **4.39** | 2.99 | 1.40 | 598.9 | 0.322 | **0.374** |
| s1.1-7B (SFT) (Muennighoff et al., 2025) | 4.26 | 2.93 | 1.32 | 593.2 | 0.317 | 0.393 |

selected categories, such global rescaling fails to close the PVC-C-PVC-VUS Gap, indicating that the trade-off is a fundamental property of the learned representations rather than a superficial scaling issue.

## L    EVALUATION ON MATH-500 BENCHMARK

To assess the robustness and generality of our framework, we replicate the full evaluation procedure on the MATH-500 benchmark (HuggingFaceH4, 2024), a well-established diagnostic suite derived from the MATH dataset.

The results largely mirror the trends observed in our main evaluation, reinforcing the robustness of our proposed metrics. OpenThinker2-7B continues to demonstrate strong introspective performance, achieving the highest PVC-VUS and C-PVC-VUS on Math-360 and ranking second on MATH-500. This consistency underscores the model's reliable self-assessment capabilities across diverse mathematical reasoning tasks.

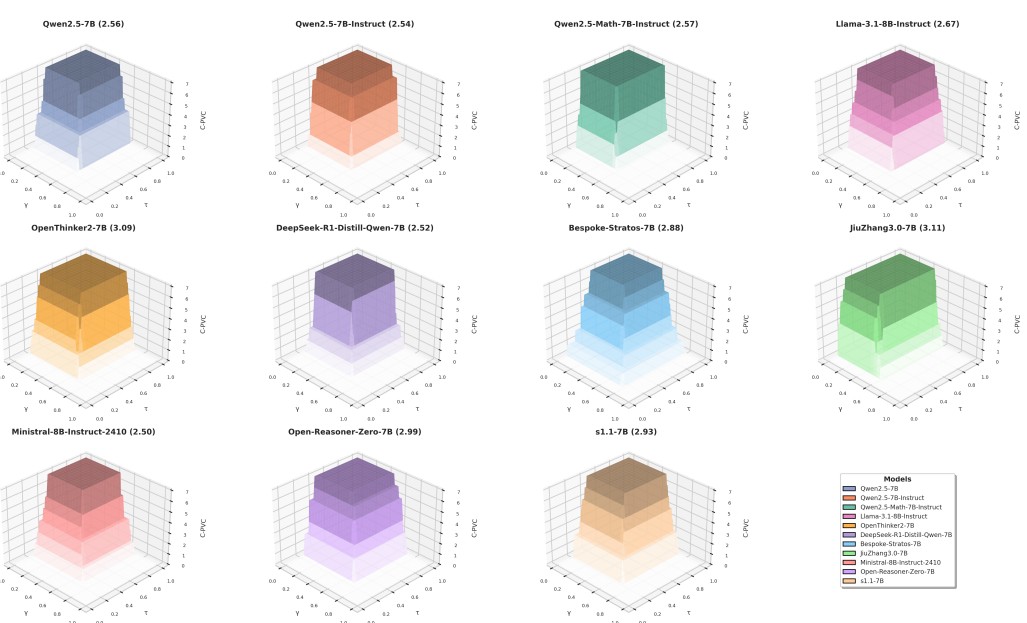

Figure 4: C-PVC Dimension on MATH-500: The x-axis ($\tau$) represents calibration tolerance, y-axis ($\gamma$) shows accuracy threshold, and z-axis displays C-PVC values.

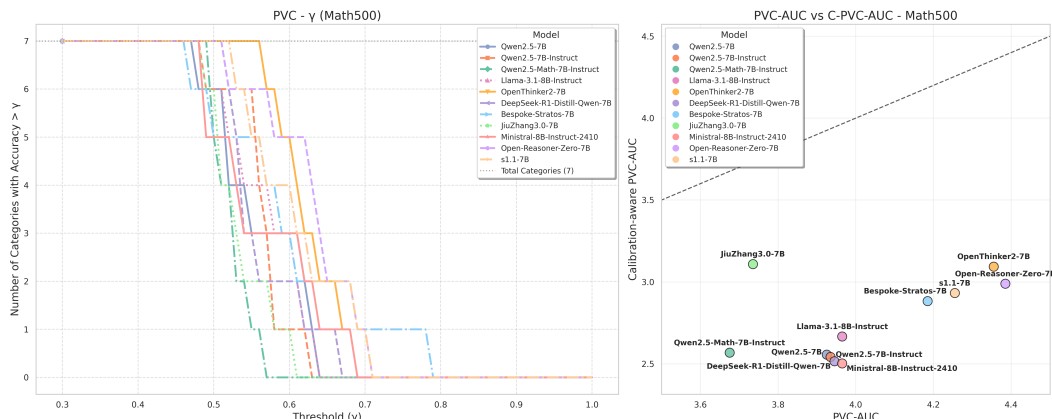

Figure 5: (a) PVC scores across accuracy thresholds ($\gamma$). (b) Comparison between PVC-VUS and C-PVC-VUS. JiuZhang3.0-7B shows the smallest gap, demonstrating superior calibration.

JiuZhang3.0-7B exhibits similarly stable behavior, consistently achieving the lowest PVC-C-PVC-VUS Gap and ECE while attaining the highest SC-VUS (610.9) on MATH-500. This superior calibration translates to higher C-PVC scores than would be expected from raw reasoning performance alone. Unlike models that systematically overestimate their correctness, JiuZhang produces conservative yet well-aligned self-assessments.

However, several ranking deviations reveal how benchmark characteristics subtly influence calibration behavior. Most notably, Llama-3.1-8B-Instruct shows degraded performance on MATH-500, with substantial drops in C-PVC-VUS, ECE, and AE rankings compared to Math-360, despite maintaining moderate overall PVC-VUS scores.

The joint calibration plot in Figure 5 shows all models falling below the diagonal, reflecting decreased C-PVC relative to PVC when calibration constraints are applied. Models such as Open-Reasoner-Zero-7B and OpenThinker2-7B position closest to the upper-right frontier, achieving optimal balance between reasoning coverage and confidence alignment. These cross-dataset results validate the stability of our evaluation framework while illuminating benchmark-specific effects.

The MATH-500 dataset, being more extensively studied, may exhibit greater overlap with certain models' training data. Nevertheless, the consistent ranking patterns in PVC and C-PVC across datasets demonstrate that these metrics provide a meaningful and generalizable approach to assessing model self-evaluation capabilities. This cross-dataset consistency further validates our framework's utility for understanding language model introspective performance.

# M    JUDGE CORRELATION STUDY

To assess the consistency of ground-truth judgments, we analyzed the correlation between model-generated self-assessments and verdicts from three reference LLMs: Claude 3.7 Sonnet, Amazon Nova Premier, and DeepSeek-R1 using the Math-360 dataset. Figure 6 illustrates two complementary metrics: Pearson correlation coefficients (linear relationship strength) and direct agreement rates (outcome-level consensus).

As shown in the heatmap (Figure 6, left), judge alignment highlights distinct evaluative signatures. **Amazon Nova Premier** demonstrates the highest consistency with target models (Avg Correlation: 0.18; Table 4), particularly with Open-Reasoner-Zero-7B ($r = 0.52$) and Qwen2.5-7B-Instruct ($r = 0.31$). This suggests that Amazon Nova Premier employs evaluation criteria congruent with the reasoning patterns of the 7–8B models under study, serving as a representative baseline.

In contrast, **Claude 3.7 Sonnet** exhibits negligible or negative correlations (Avg: -0.09), notably with Open-Reasoner-Zero-7B ($r = -0.27$). This divergence implies that Claude applies fundamentally stricter or orthogonal evaluation criteria compared to the target models. **DeepSeek-R1** occupies

a middle ground (Avg: 0.01), displaying balanced alignment without strong bias toward specific methodologies.

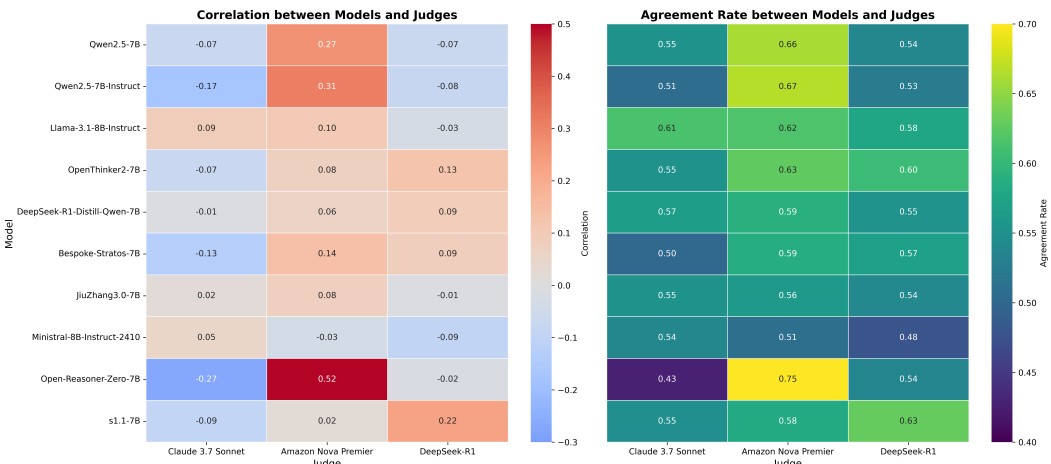

Figure 6: Left: Heatmap showing Pearson correlation between correct model self-evaluation answers and correct judge answers; Right: Heatmap showing direct agreement percentage between model self-evaluation answers and judge answers

The agreement rate analysis (Figure 6, right) corroborates these trends. Amazon Nova Premier maintains the highest average agreement (0.61), peaking at 0.75 with Open-Reasoner-Zero-7B. Conversely, Claude 3.7 Sonnet shows the lowest agreement (0.53), reinforcing the interpretation of its stringent assessment standards. Notably, the overall accuracy of models (e.g., OpenThinker2: $\approx 0.65$) often exceeds outcome-specific agreement with individual judges. This discrepancy highlights that single judges may have non-overlapping error modes. Consequently, integrating diverse perspectives—combining the strictness of Claude with the alignment of Nova—is essential to mitigate individual judge bias and ensure a robust evaluation of self-reflection capabilities.

In conclusion, the observed divergence in evaluative strictness—ranging from the high alignment of Amazon Nova to the stringent criteria of Claude 3.7—substantiates the necessity of a multi-judge protocol. Reliance on a single model risks introducing systematic bias, whereas our ensemble approach integrates these orthogonal perspectives to establish a consensus-based ground truth. This methodology ensures that our PVC and C-PVC measurements reflect robust self-evaluation capabilities rather than artifacts of a specific judge's preference.

| LLM Judge | Correlation | Agreement rate |
|---|---|---|
| Claude 3.7 Sonnet | -0.09 | 0.53 |
| Amazon Nova Premier | 0.18 | 0.61 |
| DeepSeek - R1 | 0.01 | 0.55 |

Table 4: Overall pearson correlation and agreement rate between judge and model

# N    DECOUPLING OF REASONING AND SELF-EVALUATION

A fundamental question in the study of metacognition is the degree to which self-evaluation capabilities are decoupled from primary reasoning skills. To determine whether self-assessment is merely a byproduct of generation performance or a distinct competency, we analyzed the correlation between

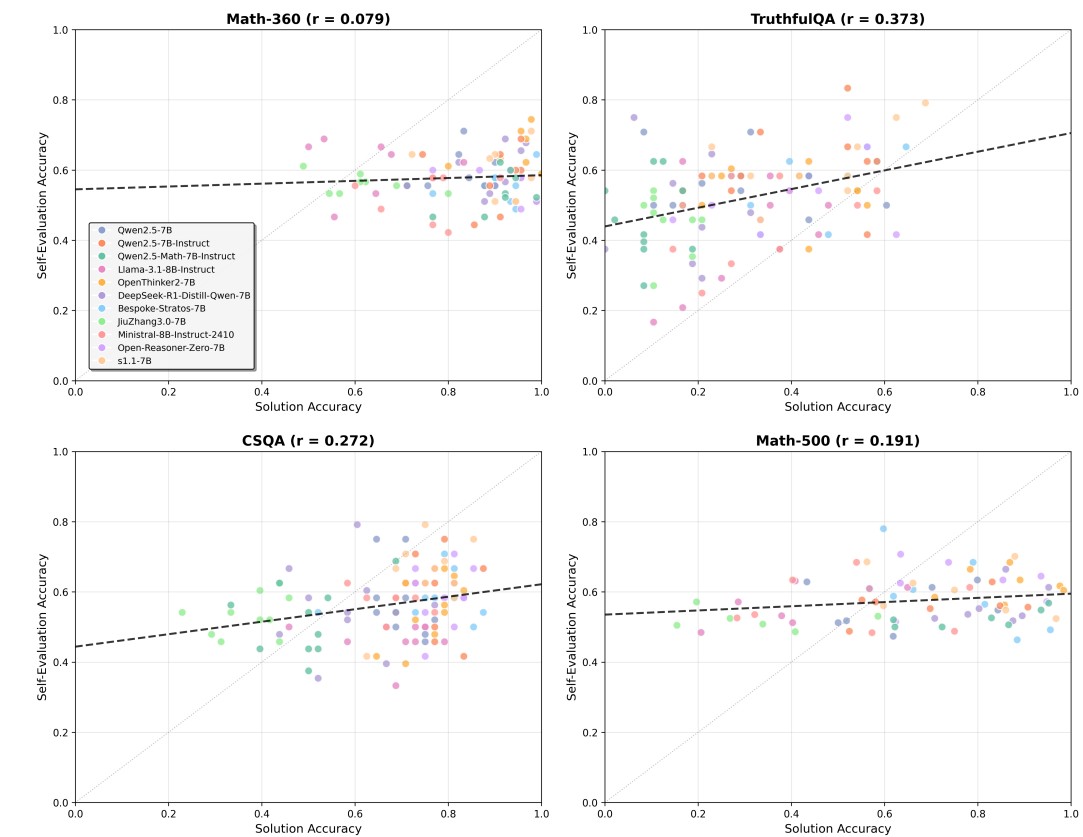

Figure 7: Correlation between Solution Accuracy (x-axis) and Self-Evaluation Accuracy (y-axis) across four benchmarks. Each point represents a specific category for a given model. The low Pearson correlation coefficients ($r$) indicate that generation and evaluation are distinct capabilities.

Total Solution Accuracy (SA) and Total Self-Evaluation Accuracy (SEA) at the category level across all 11 models.

As illustrated in Figure 7, we observe a weak correlation between generation correctness and evaluation capability across all datasets. In the domain of mathematical reasoning, this dissociation is particularly pronounced; the Pearson correlation is negligible on Math-360 ($r = 0.079$) and remains weak on MATH-500 ($r = 0.191$). This statistical independence indicates that a model's proficiency in solving mathematical problems does not guarantee its capacity to discriminate between superior and inferior solutions.

Table 5 provides quantitative evidence of this decoupling. On MATH-500, for instance, Open-Reasoner-Zero-7B achieves a Solution Accuracy (SA) of 76.6%, which is significantly lower than OpenThinker2-7B's 86.7%. However, Open-Reasoner-Zero-7B exhibits a higher Self-Evaluation Accuracy (SEA) of 62.6% compared to OpenThinker2-7B's 62.2%. Similarly, on TruthfulQA, s1.1-7B records a lower SA (44.2%) than Bespoke-Stratos-7B (45.2%), yet demonstrates a vastly superior SEA (62.1% vs. 56.7%). These inversions—where lower reasoning performance coincides with higher evaluation capability—highlight a functional dissociation between generative proficiency and evaluative reliability, indicating that high reasoning performance is not a sufficient condition for accurate self-assessment.

Consequently, these findings suggest that self-evaluation operates as a distinct metacognitive skill, structurally independent of raw reasoning generation. Metrics like C-PVC thus provide unique, orthogonal insights into model reliability that cannot be captured by standard accuracy benchmarks alone, validating the necessity of our specialized evaluation framework.

Table 5: Comparison of Total Solution Accuracy (SA) vs. Total Self-Evaluation Accuracy (SEA) across four benchmarks. All values are reported in percentages (%). The specific cases where models with lower SA achieve higher SEA (e.g., Open-Reasoner vs. OpenThinker2 on MATH-500) quantitatively substantiate that self-evaluation is an orthogonal skill.

| Model | Math-360 | | TruthfulQA | | CSQA | | MATH-500 | |
|---|---|---|---|---|---|---|---|---|
| | SA (%) | SEA (%) | SA (%) | SEA (%) | SA (%) | SEA (%) | SA (%) | SEA (%) |
| Qwen2.5-7B | 83.1 | 59.7 | 29.2 | 56.5 | 70.0 | 57.1 | 63.0 | 56.1 |
| Qwen2.5-7B-Instruct | 89.6 | 58.1 | 40.2 | 61.3 | 78.7 | 60.4 | 70.6 | 56.2 |
| Qwen2.5-Math-7B-Instruct | 90.8 | 53.9 | 9.4 | 46.3 | 49.4 | 51.7 | 79.2 | 52.4 |
| Llama-3.1-8B-Instruct | 63.9 | 60.6 | 30.4 | 41.3 | 68.5 | 46.7 | 41.4 | 56.5 |
| OpenThinker2-7B | **94.7** | **64.9** | 39.0 | 54.4 | 75.8 | 54.6 | **86.7** | 62.2 |
| DeepSeek-R1-Distill | 91.8 | 59.6 | 17.5 | 50.0 | 57.5 | 55.0 | 83.9 | 56.3 |
| Bespoke-Stratos-7B | 90.7 | 55.6 | **45.2** | 56.7 | 77.3 | 58.8 | 76.0 | 59.7 |
| JiuZhang3.0-7B | 61.7 | 56.1 | 13.8 | 45.4 | 37.1 | 53.3 | 36.0 | 53.4 |
| Ministral-8B-Instruct | 75.7 | 52.8 | 33.3 | 45.8 | 68.8 | 57.9 | 52.2 | 56.6 |
| Open-Reasoner-Zero-7B | 91.5 | 56.4 | 39.4 | 54.6 | **79.0** | 58.8 | 76.6 | **62.6** |
| s1.1-7B | 90.7 | 59.3 | 44.2 | **62.1** | 75.6 | **65.2** | 75.4 | 60.7 |

# O    ON THE INSUFFICIENCY OF RANK-BASED METRICS FOR SELF-EVALUATION ASSESSMENT

## O.1    THE ISOTONIC INVARIANCE OF AUROC

A natural question in the evaluation of probabilistic reasoning systems is why established discriminative metrics, specifically the Area Under the Receiver Operating Characteristic curve (AUROC), are insufficient for quantifying the reliability of self-reflection. The fundamental limitation of AUROC in this context stems from its property of isotonic invariance, which renders it blind to the semantic meaning of confidence scores. Formally, let $\mathcal{D}$ be a distribution over inputs $\mathcal{X}$ and binary labels $\mathcal{Y} \in \{0, 1\}$. For a probabilistic predictor $f : \mathcal{X} \to [0, 1]$, the AUROC can be interpreted probabilistically as the likelihood that a randomly selected positive instance $x^+$ is assigned a higher confidence score than a randomly selected negative instance $x^-$:

$$\text{AUROC}(f) = \mathbb{P}_{(x^+, x^-) \sim \mathcal{D}^2} \left( f(x^+) > f(x^-) \right) \tag{8}$$

This definition depends exclusively on the ordinal ranking of the predictions, independent of their absolute magnitudes. Consequently, for any strictly increasing monotonic transformation function $g : [0, 1] \to [0, 1]$, the AUROC remains invariant:

$$\text{AUROC}(f) = \text{AUROC}(g \circ f) \tag{9}$$

While this property is advantageous for tasks requiring only relative discrimination, it is theoretically inadequate for autonomous systems where the absolute value of $f(x)$, typically the reported confidence, must serve as a proxy for the true probability of correctness.

## O.2    THE CALIBRATION BLINDNESS PITFALL

The invariance described above leads to a critical failure mode we term Calibration Blindness. A model may exhibit perfect discriminative power (ranking positive instances higher than negative ones) while being dangerously miscalibrated.

Consider two hypothetical predictors, $f_A$ (Well-Calibrated) and $f_B$ (Overconfident), evaluating a binary classification task where the true label for $x_1$ is 1 and for $x_2$ is 0.

Predictor $f_A$ assigns scores $f_A(x_1) = 0.6$ and $f_A(x_2) = 0.4$. The ranking is correct ($0.6 > 0.4$), yielding an AUROC of 1.0. The scores reasonably reflect the inherent uncertainty of the task.

Conversely, Predictor $f_B$ assigns scores $f_B(x_1) = 0.9999$ and $f_B(x_2) = 0.9998$. Since the ranking is preserved ($0.9999 > 0.9998$), the AUROC remains 1.0. However, $f_B$ implies a probability of error of roughly $10^{-4}$ for the incorrect instance $x_2$, whereas the actual empirical error is 1.0. An evaluation

metric based solely on AUROC would deem $f_A$ and $f_B$ equivalent. However, deploying $f_B$ in a high-stakes environment would result in a silent failure: the system would proceed with decisive action based on a hallucinated certainty, bypassing any uncertainty-based safety thresholds.

### O.3 THE C-PVC RESOLUTION

The Calibration-aware Probabilistic VC (C-PVC) dimension provides a principled remedy to the fundamental inadequacy of AUROC by introducing a metric that is explicitly sensitive to the absolute values of confidence scores. While AUROC depends solely on the ordinal ranking of predictions, C-PVC imposes structural constraints that require predicted confidences to faithfully reflect empirical correctness. To recall, for a set to be shattered under the C-PVC framework, a predictor $f$ with an associated confidence estimator $\hat{p}$ must satisfy not only the accuracy threshold $\gamma$ but also a bounded calibration error:

$$\left| \hat{p}(x) - \mathbb{E}\big[\mathbf{1}_{\{f(x)=y\}}\big] \right| \leq \tau.$$

Applying this condition to the overconfident predictor $f_B$ from the earlier example reveals an immediate violation. For the negative instance $x_2$, the reported confidence is $\hat{p}(x_2) = 0.9998$, whereas the ground-truth realization is 0. The corresponding calibration error is

$$\left| 0.9998 - 0 \right| \approx 1.0,$$

which exceeds any reasonable tolerance (e.g., $\tau = 0.1$). Thus, although AUROC assigns a perfect score to the overconfident predictor, the C-PVC framework rightfully rejects it: the predictor fails to shatter the set under the calibration constraint. By enforcing that the distribution of reported confidences structurally matches the empirical distribution of correctness, C-PVC distinguishes between a model that merely orders predictions correctly and one that possesses a trustworthy understanding of its own uncertainty. In this way, C-PVC serves not as a purely discriminative metric, but as a rigorous measure of calibrated self-reflection, ensuring that a model's confidence is meaningfully aligned with its true limitations.

## P COMPARATIVE ADVANTAGES OF THE PVC FRAMEWORK

While standard metrics such as Expected Calibration Error (ECE) and Area Under the ROC Curve (AUROC) provide valuable insights into model performance, they exhibit fundamental limitations when assessing the structural reliability of self-reflection. The Probabilistic VC (PVC) framework addresses these shortcomings through its inherent calibration awareness and, critically, its categorical decomposability.

### P.1 BEYOND RANKING AND AVERAGES

As detailed in Appendix O, rank-based metrics like AUROC suffer from isotonic invariance, rendering them blind to calibration errors. Similarly, ECE provides only a global average, where opposing errors across domains can cancel each other out, obscuring specific structural failures.

### P.2 STRUCTURAL DECOMPOSABILITY VIA DISJOINT PARTITIONING

A distinct and critical advantage of the PVC framework is its mathematical formulation based on the cardinality of shattered sets, which enables linear decomposability. This framework operates under the premise that the comprehensive input space of reasoning tasks, denoted as $\mathcal{X}_{\text{total}}$, admits a disjoint partition into orthogonal semantic domains (e.g., $\mathcal{X}_{\text{Math}}$, $\mathcal{X}_{\text{Logic}}$, $\mathcal{X}_{\text{Fact}}$). Unlike the current experimental scope which is necessarily finite, this theoretical formulation presupposes that such partitioning can be extended to cover the full spectrum of general intelligence. Under this assumption of orthogonality, the total capacity of the model space $\mathcal{F}$ over the union of these domains is strictly additive. Crucially, this additivity holds for both the expressive capacity (PVC) and the calibrated capacity (C-PVC). Unlike non-linear metrics such as AUC, where the integral over a combined domain cannot be derived from component scores, the dimensions over a disjoint union $\mathcal{X}_{\text{total}} = \bigcup_i \mathcal{X}_i$ satisfy:

$$\text{PVC}_{\text{Total}}(\mathcal{F}) = \sum_i \text{PVC}_{\mathcal{X}_i}(\mathcal{F})$$

$$\text{C-PVC}_{\text{Total}}(\mathcal{F}) = \sum_i \text{C-PVC}_{\mathcal{X}_i}(\mathcal{F}) \tag{10}$$

This dual equality implies that the framework supports **modular extensibility**: as the evaluation expands to broader contexts (e.g., adding a new domain $\mathcal{X}_{\text{Coding}}$), the global reliability score can be updated simply by adding the marginal capacity of the new domain, without normalizing or re-evaluating existing scores. This allows researchers to mathematically decompose not just the model's raw reasoning power, but also its trustworthiness into constituent parts. Consequently, the PVC framework provides a rigorous, scalable balance sheet of reliability, capable of mapping the hierarchical knowledge structure of Large Language Models far beyond the constraints of limited benchmarks.

## Q  EXTENDED RELATED WORKS

**Evolution of self-reflection.** The study of self-reflection traces its lineage from educational psychology frameworks (Brown, 2010; Govaerts et al., 2012; Johnston et al., 2005; Leijen et al., 2009; Nelson & Freier, 2008; Palinscar & Brown, 1984; Reznitskaya et al., 2012; Tseng & Bryant, 2013; Webb et al., 2013) to modern LLM-specific methodologies (Shi, 2019; Bentvelzen et al., 2022). In this domain, self-reflection denotes the mechanism for examining internal reasoning, while self-evaluation quantifies confidence. Contemporary approaches such as Reflexion (Shinn et al., 2023), Self-Refine (Madaan et al., 2023), CRITIC (Pan et al., 2023), and $R^3$ (Wang et al., 2025) implement these concepts via iterative refinement loops. Theoretical efforts, including Metacognitive Prompting (Toy et al., 2024) and metacognitive language models (Wang & Zhao, 2023), have attempted to formalize these processes, laying the groundwork for our complexity-theoretic analysis.

**Probabilistic learning foundations.** While extensions of VC theory to probabilistic predictors exist in limited contexts (Klesk, 2012), they have not been systematically applied to neural language models. Our framework bridges this gap by leveraging fat-shattering dimension theory (Mendelson & Vershynin, 2003; Anthony & Bartlett, 2009; Telgarsky, 2017; Bartlett et al., 2019; Colomboni et al., 2025). We extend these foundational results to establish a theoretical link between classical learning guarantees and the probabilistic outputs of modern LLMs.

**Calibration and uncertainty quantification.** Ensuring reliability involves calibration techniques ranging from post-hoc temperature scaling (Platt et al., 1999) and conformal prediction (Angelopoulos & Bates, 2021) to uncertainty quantification methods like deep ensembles (Lakshminarayanan et al., 2016) and Bayesian approximations (Gal & Ghahramani, 2015; Woo, 2023; 2022), with recent works proposing balanced entropy measures to better capture the information trade-off in probabilistic predictions Woo (2023). Our work integrates these calibration concepts into a capacity measurement framework, distinct from prior works that often treat calibration in isolation from expressive power.

**Alignment and reasoning optimization.** Recent advances in model training, including RLHF (Ouyang et al., 2022), Direct Preference Optimization (DPO) (Rafailov et al., 2023), Contrastive Preference Learning (Hejna et al., 2023), and self-rewarding mechanisms (Yuan et al., 2024), have significantly enhanced LLM capabilities (Swamy et al., 2025). Specialized variants like Group Relative Policy Optimization (GRPO) (Shao et al., 2024) explicitly target reasoning tasks. Our study complements these optimization techniques by providing a principled metric (C-PVC) to evaluate whether such training improves actual introspective reliability or merely suppresses calibration errors.

**Synthesis.** Collectively, these research streams converge in our work. We leverage classical VC theory to construct a rigorous framework for evaluating self-reflection, addressing the critical challenge of calibration in deep learning. This approach provides a theoretical foundation for understanding how diverse training methodologies influence a model's ability to reliably assess its own reasoning, effectively bridging the gap between theoretical learning guarantees and practical LLM development.

## R  DETAILED EXPERIMENTAL METHODOLOGY

This section elaborates on our empirical approach to measuring PVC dimensions in language models, providing implementation details for the framework introduced in Section 3.3.

## R.1 MEASUREMENT PROTOCOL

Our empirical evaluation follows a three-stage protocol designed to measure a model's capacity for reasoning self-assessment while avoiding information leakage. The process is illustrated in Figure 1.

**Stage 1: Answer generation.** For each problem instance, the model generates two independent solutions using distinct decoding trajectories:

- A solution that follows standard, step-by-step reasoning patterns, and
- A solution that explores a non-standard, potentially more intuitive or compact reasoning path.

These variants ensure diversity in reasoning quality while remaining model-generated.

**Stage 2: Self-evaluation.** The model is then presented with both candidate solutions in randomized order. It must choose the answer it believes to be more correct and assign a confidence score to its choice. Notably, the model is not given access to the ground-truth answer at any point, ensuring that this choice reflects genuine internal evaluation, not supervised feedback.

**Stage 3: External judging.** To determine the ground-truth correctness, a separate ensemble of larger language models—acting as external judges—is given access to the same pair of solutions, along with the correct final answer. The ensemble selects the objectively superior solution, serving as the gold label for evaluating self-reflection accuracy. This design ensures that the evaluation signal is (i) grounded in correctness, (ii) independent of the model being evaluated, and (iii) robust to individual model biases.

## R.2 HYPERPARAMETERS

The decoding and evaluation configuration reported below was applied identically to all eleven models examined in this study—QWEN2.5-7B (Yang et al., 2024a), QWEN2.5-7B-INSTRUCT (Yang et al., 2024a), QWEN2.5-MATH-7B-INSTRUCT (Yang et al., 2024b), DEEPSEEK-R1-DISTILL-QWEN-7B (Guo et al., 2025), LLAMA-3.1-8B-INSTRUCT (Meta AI, 2024), OPENTHINKER2-7B (Open-Thoughts, 2023), BESPOKE-STRATOS-7B (Bespoke Labs, 2023), JIUZHANG3.0-7B (Zhou et al., 2024), MINISTRAL-8B-INSTRUCT-2410 (Mistral AI, 2024), OPEN-REASONER-ZERO-7B (Hu et al., 2025), and S1.1-7B (Muennighoff et al., 2025). The same parameter settings were used for every query sent to the judge ensemble.

Table 6: Decoding and evaluation settings

| Parameter | Temperature | Top-P | Max tokens | Judge ensemble |
|---|---|---|---|---|
| Value | 0.7 | 0.9 | 4096 | Claude 3.7 Sonnet (Anthropic, 2025); Amazon Nova Premier (Intelligence, 2024); DeepSeek-R1 (Guo et al., 2025) |

## R.3 MATH-360 BENCHMARK DATASET CONSTRUCTION

To evaluate mathematical reasoning capabilities more effectively, we developed a new benchmark that addresses one potential limitation of some existing datasets: the possibility of data contamination. Certain prior benchmarks—particularly those derived from widely available competition problems, textbooks, or online resources—may have some degree of overlap with pretraining corpora of large language models, which could potentially influence performance measurements. Our benchmark was developed with attention to originality and domain diversity, aiming to provide a complementary evaluation resource that helps assess reasoning abilities while reducing the likelihood of familiarity effects.

**Taxonomic coverage.** Table 7 presents the taxonomy of mathematical reasoning categories and sub-categories included in our benchmark. The dataset spans eight core domains—Arithmetic, Algebra, Calculus, Geometry, Number Theory, Combinatorics, Statistics, and Linear Algebra—each subdivided into five foundational subtopics. This taxonomy reflects the structure of standard mathematics curricula and facilitates interpretable analysis across well-defined conceptual boundaries.

Table 7: Taxonomy of mathematical reasoning categories in our benchmark.

| Category | Subcategories |
|---|---|
| Arithmetic | Basic Operations, Fractions, Percentages, Numerical Approximation, Order of Operations |
| Algebra | Equations, Inequalities, Polynomials, Functions, Systems of Equations |
| Calculus | Differentiation, Integration, Limits, Series, Applications |
| Geometry | Plane Geometry, Coordinate Geometry, Transformations, Mensuration, Trigonometry |
| Number Theory | Divisibility, Modular Arithmetic, Primes, Diophantine Equations, Number Sequences |
| Combinatorics | Counting Principles, Permutations, Combinations, Probability, Recursion |
| Statistics | Descriptive Statistics, Distributions, Hypothesis Testing, Regression, Bayesian Inference |
| Linear Algebra | Matrices, Determinants, Vector Spaces, Eigenvalues, Linear Transformations |

**Problem stratification and complexity.**  Each subcategory contains five problems rigorously stratified across difficulty levels (Easy, Medium, Hard), yielding a total of 360 problems. Crucially, while the Easy tier validates fundamental competencies, the Hard tier incorporates complex, multi-step reasoning tasks. This broad difficulty spectrum is designed to stress-test the reasoning limits of 7–8B models. We note that the examples in Table 8 are selected for brevity and illustrative clarity; the actual dataset includes problems requiring extensive intermediate derivations to prevent ceiling effects.

**Representative examples.**  Table 8 provides one representative problem per subcategory, highlighting the range and depth of reasoning skills required. Problems are designed to elicit multi-step thinking, abstraction, and formal manipulation, rather than direct recall or pattern matching. For instance, problems in combinatorics test combinatorial enumeration and recurrence, while those in number theory evaluate modular reasoning and structural properties of integers.

Taken together, the benchmark offers a principled and contamination-resistant testbed for evaluating mathematical reasoning in language models. It supports both overall capability assessment and fine-grained diagnosis of strengths and weaknesses across mathematical domains.

### R.4   CONSTRUCTION OF NON-MATHEMATICAL BENCHMARKS

To investigate the generalizability of our framework beyond mathematical reasoning, we incorporated two established benchmarks: TruthfulQA (Lin et al., 2022) and CommonsenseQA (Talmor et al., 2019). However, utilizing these datasets in their raw form presents challenges for our category-level analysis (PVC/C-PVC) due to inherent class imbalances; the original datasets contain a vast number of questions with highly uneven distributions across different semantic topics.

To ensure statistical stability and prevent evaluation bias toward dominant categories, we implemented a rigorous curation protocol to align with the balanced structure of Math-360:

1. **Category selection:** For each benchmark, we identified the top-10 major categories based on semantic labels to ensure broad coverage of the domain.

2. **Uniform balancing constraint:** To maintain consistency, we identified the category with the minimum number of available samples among the top 10. This count served as the limiting factor for our stratified sampling.

3. **Stratified sampling:** Based on this constraint, we randomly sampled exactly 24 distinct questions from each of the 10 categories.

This process yielded a standardized subset of 240 questions per benchmark (10 categories $\times$ 24 questions). By enforcing this uniform distribution, we ensure that our PVC and C-PVC estimates reflect the model's true capability across diverse reasoning patterns rather than being skewed by the volume of data in specific sub-domains.

Table 8: Representative problems from each subcategory in our mathematical reasoning benchmark 360.

| Category | Subcategory | Example Problem |
|---|---|---|
| Arithmetic | Basic Operations | Calculate $238 + 149$. |
| | Fractions | What is $\frac{2}{5} + \frac{1}{5}$? |
| | Percentages | What is $25\%$ of $80$? |
| | Numerical Approximation | Round $47.68$ to the nearest whole number. |
| | Order of Operations | Calculate $3 + 4 \times 2$. |
| Algebra | Equations | Solve for $x$: $x + 5 = 12$. |
| | Inequalities | Solve: $x + 3 > 7$. |
| | Polynomials | Simplify: $(3x^2 + 2x) + (4x^2 - 5x + 1)$. |
| | Functions | If $f(x) = 2x + 3$, find $f(4)$. |
| | Systems of Equations | Solve the system: $x + y = 5, x - y = 1$. |
| Calculus | Differentiation | Find the derivative of $f(x) = 3x^2 + 2x - 5$. |
| | Integration | Find $\int (3x^2 + 2)\, dx$. |
| | Limits | Evaluate $\lim_{x \to 3}(x^2 - 4)$. |
| | Series | Find the sum of the first 10 terms of the arithmetic sequence with $a_1 = 3$ and $d = 4$. |
| | Applications | Find the maximum value of $f(x) = -x^2 + 6x - 5$ on the interval $[0, 5]$. |
| Geometry | Plane Geometry | Find the area of a rectangle with length 8 cm and width 5 cm. |
| | Coordinate Geometry | Find the distance between the points $(3, 4)$ and $(6, 8)$. |
| | Transformations | Reflect the point $(3, 5)$ across the $x$-axis. |
| | Mensuration | Find the circumference of a circle with radius 5 cm. |
| | Trigonometry | Find $\sin(30°)$. |
| Number Theory | Divisibility | Determine whether $156$ is divisible by $4$. |
| | Modular Arithmetic | Calculate $17 \pmod 5$. |
| | Primes | List all prime numbers less than 20. |
| | Diophantine Equations | Find all integer solutions to $x + y = 10$. |
| | Number Sequences | Find the next number in the sequence: $3, 7, 11, 15, \ldots$ |
| Combinatorics | Counting Principles | How many different 3-digit numbers can be formed using the digits $1, 2, 3, 4, 5$ without repetition? |
| | Permutations | How many permutations can be formed using all the letters of the word 'MATH'? |
| | Combinations | How many ways are there to select 3 books from a shelf containing 7 different books? |
| | Probability | A fair die is rolled. What is the probability of getting a number greater than 4? |
| | Recursion | Find the 6th term in the Fibonacci sequence, where $F_1 = 1, F_2 = 1$, and $F_n = F_{n-1} + F_{n-2}$ for $n \geq 3$. |
| Statistics | Descriptive Statistics | Find the mean of the numbers $4, 7, 9, 3$, and $2$. |
| | Distributions | If $X$ follows a normal distribution with mean 10 and standard deviation 2, what is $P(X > 12)$? |
| | Hypothesis Testing | In a hypothesis test, what is the meaning of a Type I error? |
| | Regression | In a linear regression model, what does the slope coefficient represent? |
| | Bayesian Inference | State Bayes' theorem. |
| Linear Algebra | Matrices | Calculate the sum of matrices $A = \begin{bmatrix} 3 & 5 \\ 1 & 2 \end{bmatrix}$ and $B = \begin{bmatrix} 4 & 0 \\ 6 & 3 \end{bmatrix}$. |
| | Determinants | Calculate the determinant of the matrix $A = \begin{bmatrix} 3 & 4 \\ 1 & 2 \end{bmatrix}$. |
| | Vector Spaces | Determine whether the set of vectors $\{(1, 0, 0), (0, 1, 0), (0, 0, 1)\}$ forms a basis for $\mathbb{R}^3$. |
| | Eigenvalues | Find the eigenvalues of the matrix $A = \begin{bmatrix} 2 & 0 \\ 0 & 3 \end{bmatrix}$. |
| | Linear Transformations | Let $T : \mathbb{R}^2 \to \mathbb{R}^2$ be defined by $T(x, y) = (2x, 3y)$. Find $T(1, 2)$. |

## R.5 EXPERIMENTAL PROMPTS

Our measurement procedure operationalizes the theoretical concept of PVC dimension through a three-stage process.

### R.5.1 FIRST SOLUTION GENERATION

The target model generates an expert-level solution using the following prompt:

> **System:** You are an expert mathematics educator with deep knowledge across all mathematical domains. Your task is to solve a math problem with exceptional clarity and accuracy.

> **User:** Please solve the following mathematical problem step by step, ensuring complete accuracy:
> PROBLEM: {problem}
> Provide a detailed solution that demonstrates expert-level understanding.
> Include:
> 1. Clear explanation of your approach
> 2. Each step fully worked out
> 3. Proper mathematical notation
> 4. Verification of your answer
> Make sure your final answer is clearly indicated and correct.

### R.5.2 SECOND SOLUTION GENERATION

The model generates an alternative solution using:

> **System:** You are a brilliant mathematics educator known for your creative problem-solving approaches. Your task is to solve a math problem using your new methods.

> **User:** Please solve the following mathematical problem:
> PROBLEM: {problem}
> Provide an extremely detailed solution that demonstrates understanding.
> Include:
> 1. A clear explanation of your approach using an alternative and creative solution path
> 2. Each step fully worked out with mathematical reasoning
> 3. Proper mathematical notation and formalism
> 4. If applicable, connect this problem to broader mathematical concepts or real-world applications
> Make your solution as thorough and educational as possible, showing multiple ways to think about and verify the solution. Focus on approaches that demonstrate mathematical insight and elegance.

This design choice—requesting alternative approaches rather than explicitly introducing errors—allows for natural quality variation while avoiding artificially degraded solutions that would be too obviously flawed.

### R.5.3 SELF-EVALUATION FOR MATH DATASET

The model evaluates both solutions (presented in randomized order), identifies which it believes is superior, and provides a confidence score (0-100%). The evaluation prompt follows:

> **System:** You are an expert mathematics evaluator tasked with comparing two solutions to the same problem.

> **User:** Below are two different solutions to the same mathematical problem. Your task is to carefully evaluate both solutions and determine which one is superior in terms of correctness, clarity, and mathematical insight.
> PROBLEM: {problem}
> SOLUTION A: {solution_a}
> SOLUTION B: {solution_b}
> Please compare these solutions carefully and respond with:
> 1. Which solution is better (A or B)
> 2. Your confidence in this judgment (0-100%)
> 3. A brief explanation of why one solution is superior to the other
> Format your response as:
> Better solution: [A or B]
> Confidence: [0-100%]
> Explanation: [Your analysis]

## S EXAMPLES

The following examples are provided for illustrative purposes only, to give readers a concrete understanding of how models perform self-evaluation in our framework. While our quantitative results in Section 5 comprehensively evaluate model performance through PVC and C-PVC metrics, these examples offer qualitative insights into the self-evaluation process. These examples illustrate when the model exhibits accurate introspective judgment, and when it fails due to overconfidence or conceptual misunderstanding.

### S.1 EXAMPLES OF SELF-EVALUATION BEHAVIOR

We present two representative cases—one demonstrating successful self-evaluation and another showing flawed assessment—to highlight the different patterns of reasoning that emerge during

self-reflection. These examples are anonymized and selected to illustrate typical behaviors observed across multiple models, rather than to evaluate any specific model's performance.

**Case 1: Correct Self-Evaluation.** In this example, the model is asked to compute the inverse of the function $f(x) = \frac{3x-2}{x+1}$. It generates two answers: Solution A follows a rigorous, textbook-style derivation, while Solution B uses a creative algebraic reformulation. Both arrive at algebraically equivalent forms, though Solution B makes minor verification errors. The model correctly selects Solution A and assigns a confidence score of 90. This choice aligns with the external judge's label, and the model's justification reflects meaningful understanding of correctness. Thus, both its selection and confidence are well-calibrated, yielding high values in PVC and C-PVC.

**Case 2: Flawed Self-Evaluation.** Here, the model is asked whether $f(x) = |x^2 - 4|$ is one-to-one, and if not, to restrict the domain and compute its inverse. Solution A correctly identifies the need for domain restriction (e.g., $x \geq 2$) and derives the inverse. In contrast, Solution B makes multiple conceptual errors, such as misinterpreting the absolute value function as a square root and incorrectly restricting the domain. Despite these flaws, the model selects Solution B with a confidence score of 80, justifying its choice based on surface-level clarity. This leads to a self-reflection error: the model is confident in a mathematically invalid solution. While PVC remains high (a choice was made), C-PVC is low due to confidence miscalibration.

Table 9: Comparison of two model self-evaluation behaviors.

| Aspect | Case 1: Rational Function Inversion | Case 2: One-to-One Determination |
|---|---|---|
| Task Type | Inverse function derivation via algebraic manipulation | One-to-one check and domain-restricted inverse |
| Judge Ensemble Correct Answer | Solution A | Solution A |
| Model's Selected Answer | Solution A | Solution B |
| Confidence Score | 90 | 80 |
| Model's Justification | Emphasized rigor and step-by-step reasoning | Praised perceived clarity despite flawed math |
| Actual Reasoning Errors | None; correct and calibrated | Misinterpreted absolute value; incorrect domain |

**Discussion.** These examples highlight the need for introspective evaluation beyond accuracy. While both cases involve the model choosing between two self-generated solutions, only one reflects proper alignment between correctness and confidence. Our proposed PVC and C-PVC metrics allow us to distinguish not just whether a model chooses the right answer, but whether it knows it is right. Such analysis reveals that some models (e.g., RLHF-finetuned) may appear confident while being systematically miscalibrated, whereas others (e.g., distilled or conservative models) exhibit more cautious but reliable introspection.

## S.2 Case 1: Correct Self-Evaluation

> **User:** Find the inverse function of $f(x) = \frac{3x-2}{x+1}$.

**Solution A:** Step 1: Understanding the Problem

To find the inverse function of $f(x)$, we need to express the original function in a way that allows us to easily swap the roles of $x$ and $y$. The function given is $f(x) = \frac{3x-2}{x+1}$.

Step 2: Setting Up the Inverse Function

To find the inverse function, we'll start by letting $y = f(x)$, which gives us the equation:

$y = \frac{3x-2}{x+1}$

Next, we'll switch the roles of $x$ and $y$ to begin solving for the inverse function. This means replacing $y$ with $x$ and replacing $x$ with $y$ in the original equation.

Step 3: Swapping Variables

Swapping the variables gives us:

$x = \frac{3y-2}{y+1}$

Step 4: Solving for $y$

To solve for $y$, we'll first multiply both sides of the equation by $(y + 1)$ to get rid of the fraction:

$x(y + 1) = 3y - 2$

Step 5: Expanding and Simplifying

Now, we'll expand the left side of the equation and simplify:

$xy + x = 3y - 2$

Next, we'll move all terms involving $y$ to one side of the equation and all other terms to the other side:

$xy - 3y = -x - 2$

Step 6: Factoring and Solving

We can factor out $y$ from the terms on the left side:

$y(x - 3) = -x - 2$

To isolate $y$, we'll divide both sides of the equation by $(x - 3)$:

$y = \frac{-x-2}{x-3}$

Step 7: Writing the Inverse Function

Now that we've solved for $y$, we can write the inverse function of $f(x)$:

$f^{(-1)}(x) = \frac{-x-2}{x-3}$

Verification

To verify that this is the correct inverse function, we can check that $f(f^{(-1)}(x)) = x$ and $f^{(-1)}(f(x)) = x$.

Let's start by checking that $f(f^{(-1)}(x)) = x$:

$f(f^{(-1)}(x)) = f(\frac{-x-2}{x-3})$

Substituting this into the original function $f(x) = \frac{3x-2}{x+1}$, we get:

$f(f^{(-1)}(x)) = \frac{3(\frac{-x-2}{x-3})-2}{(\frac{-x-2}{x-3})+1}$

Simplifying this expression will show that $f(f^{(-1)}(x)) = x$, which verifies that the inverse function is correct.

Final Answer

The inverse function of $f(x) = \frac{3x-2}{x+1}$ is:

$f^{(-1)}(x) = \frac{-x-2}{x-3}$

This answer is verified through a detailed and accurate solution process.

**Solution B:** Inverse Function Solution: A Creative Approach

Step 1: Set up the function equation

We are given the function $f(x) = \frac{3x-2}{x+1}$. To find the inverse function, we'll use a substitution method. Let's start by setting $y = f(x)$:

$y = \frac{3x-2}{x+1}$

Step 2: Switch $x$ and $y$

To find the inverse, we need to switch the roles of $x$ and $y$. This means that we'll replace $x$ with $y$ and $y$ with $x$:

$x = \frac{3y-2}{y+1}$

Step 3: Solve for $y$

Now, we need to solve this new equation for $y$. Our goal is to isolate $y$ on one side of the equation. We can do this by first multiplying both sides by $y + 1$ to eliminate the fraction:

$x(y + 1) = 3y - 2$

Step 4: Distribute $x$ and simplify

Distribute $x$ to the terms inside the parentheses:

$xy + x = 3y - 2$

Now, let's move all terms involving $y$ to one side and the constant terms to the other side:

$xy - 3y = -x - 2$

Step 5: Factor out $y$ and simplify

Factor out $y$ from the terms on the left-hand side:

$y(x - 3) = -x - 2$

Step 6: Solve for $y$

To solve for $y$, divide both sides by $x - 3$. However, we must consider two cases to avoid division by zero:

Case 1: $x \neq 3$

$y = \frac{-x-2}{x-3}$

Case 2: $x = 3$ (this is a special case that we'll revisit later)

For the general case $x \neq 3$, we have found the inverse function.

Step 7: Simplify the inverse function

Let's rewrite the inverse function in a simpler form by multiplying the numerator and denominator by -1 to make the fraction more elegant:

$y = \frac{x+2}{3-x}$

This is our inverse function.

Step 8: Verify the inverse function

To verify that our inverse function is correct, we can plug it back into the original function:

$f(f^{-1}(x)) = f\left(\frac{x+2}{3-x}\right)$

$f(f^{-1}(x)) = \frac{3\left(\frac{x+2}{3-x}\right)-2}{\left(\frac{x+2}{3-x}\right)+1}$

Step 9: Simplify the expression

To simplify this expression, multiply the numerator and denominator by $(3 - x)$ to eliminate the fraction:

$f(f^{-1}(x)) = \frac{3(x+2)-2(3-x)}{(x+2)+(3-x)}$

$f(f^{-1}(x)) = \frac{3x+6-6+2x}{3}$

$f(f^{-1}(x)) = \frac{5x}{3}$

Step 10: Conclusion

We started with the function $f(x) = \frac{3x-2}{x+1}$ and found its inverse function using a creative substitution method. The inverse function is $f^{-1}(x) = \frac{x+2}{3-x}$. We verified the inverse function by plugging it back into the original function and simplifying the expression.

Broader Mathematical Concepts and Real-World Applications

Inverse functions have numerous applications in various fields, including physics, engineering, and computer science. For example, in physics, the inverse function of a displacement-time function can be used to find the velocity and acceleration of an object. In engineering, the inverse function of a stress-strain function can be used to design materials with specific properties. In computer science, inverse functions are used in algorithms for solving systems of equations and in data analysis.

Conclusion

In this solution, we demonstrated a creative approach to finding the inverse function of $f(x) = \frac{3x-2}{x+1}$ using a substitution method. We worked through each step with mathematical reasoning and proper notation, and we verified the inverse function by plugging it back into the original function. We also connected this problem to broader mathematical concepts and real-world applications, highlighting the importance of inverse functions in various fields.

**Self-Evaluation:**
**Selected Solution:** A
**Confidence Score:** 90
Upon reviewing both solutions, I noticed that Solution A is more thorough and accurate in its step-by-step process. It starts by properly setting up the inverse function, swapping the variables, and solving for y. In contrast, Solution B uses a more creative approach but lacks the necessary rigor and clarity in its solution process.
One major issue with Solution B is its failure to properly verify the inverse function. While it attempts to simplify the expression by multiplying the numerator and denominator by (3 - x), it does not actually demonstrate that this process is valid or that it correctly solves for y. Additionally, Solution B's approach is more prone to errors due to its reliance on a single substitution, whereas Solution A uses a more systematic approach that involves multiple steps.
Another issue with Solution B is its use of the special case x = 3, which is not properly addressed. While it mentions this case in passing, it does not provide a clear solution or explanation for this scenario.
In terms of broader mathematical concepts and real-world applications, both solutions could benefit from more connections to these areas. However, Solution A provides a more solid foundation in algebraic manipulation and inverse function theory, making it a more reliable choice for students and educators.
Overall, I would rate Solution A as the better of the two solutions due to its thoroughness, accuracy, and clarity. While Solution B attempts to be more creative, its flaws and omissions make it less reliable.

---

**Analysis of Self-Evaluation:**
This is an example of a flawed self-evaluation. The evaluator correctly selected Solution A as the better answer, but their reasoning contains significant errors and misunderstandings:
The evaluator claims Solution B "fails to properly verify the inverse function," but in reality, Solution B attempts verification but makes a computational error in Step 9. When simplifying $(x + 2) + (3 - x)$, the result should be 5, not 3. This leads to the incorrect final verification result of $\frac{5x}{3}$ instead of x.
The evaluator praises Solution A's verification process, but Solution A only outlines the verification without actually completing it. It merely states "Simplifying this expression will show that $f(f^{(-1)}(x)) = x$ without performing the calculations.
The evaluator mentions that Solution B has issues with addressing the special case x = 3, but this is actually a strength of Solution B, as it acknowledges domain restrictions that Solution A completely ignores.
The evaluator fails to recognize that Solution A provides its final answer as $\frac{-x-2}{x-3}$ while Solution B gives $\frac{x+2}{3-x}$. These are equivalent after multiplying numerator and denominator by -1, but the evaluator doesn't acknowledge this.
A better self-evaluation would have recognized that both solutions follow similar algebraic approaches but Solution B actually attempts (though with errors) the verification that Solution A merely mentions. It should also have noted that both solutions arrive at equivalent forms of the inverse function, with Solution B taking the extra step to present it in a slightly more elegant form.

## S.3 CASE 2: FLAWED SELF-EVALUATION

**User:** Determine whether the function $f(x) = |x^2 - 4|$ is one-to-one. If not, restrict the domain to make it one-to-one and find its inverse.

**Solution A:** To determine whether the function $f(x) = |x^2 - 4|$ is one-to-one, we need to understand what it means for a function to be one-to-one. A function f is one-to-one if it assigns distinct outputs to distinct inputs, meaning that no two different inputs produce the same output.

Let's start by analyzing the given function $f(x) = |x^2 - 4|$. This is an absolute value function composed with a quadratic function.

Step 1: Understand the nature of the quadratic function The quadratic function $x^2 - 4$ has its vertex at (0, -4) and its roots at $x = -2$ and $x = 2$. This means that the graph of this function is a parabola that opens upwards, with x-intercepts at (-2, 0) and (2, 0).

Step 2: Understand the absolute value function The absolute value function —u— returns the non-negative value of u, where u is any real number. This means that $|x^2 - 4|$ will always produce non-negative values, except when $x^2 - 4$ is negative, which occurs when $-4 < x^2 - 4 < 0$.

Step 3: Determine the behavior of the function $f(x) = |x^2 - 4|$ The graph of $f(x) = |x^2 - 4|$ consists of two branches: one above the x-axis for x ¡ -2 and one below the x-axis for x ¿ 2, and both are symmetric about the y-axis. This is becVUSe the absolute value function $|x^2 - 4|$ takes the absolute value of $x^2 - 4$, resulting in non-negative values for both branches.

Step 4: Check if the function is one-to-one To check if $f(x) = |x^2 - 4|$ is one-to-one, we need to verify if it satisfies the condition that no two different inputs produce the same output. Since the graph of $f(x) = |x^2 - 4|$ consists of two branches, one above the x-axis and one below, we can see that this function does not satisfy the one-to-one condition.

Step 5: Restrict the domain to make the function one-to-one To make $f(x) = |x^2 - 4|$ one-to-one, we need to restrict its domain to either $x \geq 2$ or $x \leq -2$. This is becVUSe restricting the domain to either of these intervals will ensure that the function has an inverse.

Step 6: Find the inverse of the function $f(x) = |x^2 - 4|$ Let's find the inverse of $f(x) = |x^2 - 4|$, restricted to $x \geq 2$.

We can start by writing $y = |x^2 - 4|$ and solving for x in terms of y. Since $y \geq 0$, we have:

$y = x^2 - 4 \ y + 4 = x^2 \ x^2 = y + 4 \ x = \pm\sqrt{(y+4)}$

However, since $x \geq 2$, we only take the positive square root:

$x = \sqrt{(y+4)}$

Substituting $y = |x^2 - 4|$ back into the equation, we get:

$x = \sqrt{(|x^2 - 4| + 4)}$

This can be simplified to:

$f^{-1}(x) = \sqrt{(x+4)}$

Step 7: Verify the inverse To verify that $f^{-1}(x) = \sqrt{(x+4)}$ is the inverse of $f(x) = |x^2 - 4|$, we need to check that $f(f^{-1}(x)) = x$ and $f^{-1}(f(x)) = x$.

Substituting $f^{-1}(x) = \sqrt{(x+4)}$ into $f(x) = |x^2 - 4|$, we get:

$f(f^{-1}(x)) = f(\sqrt{(x+4)}) = |(\sqrt{(x+4)})^2 - 4| = |x + 4 - 4| = |x| = x$

Similarly, substituting $f(x) = |x^2 - 4|$ into $f^{-1}(x) = \sqrt{(x+4)}$, we get:

$f^{-1}(f(x)) = \sqrt{(|x^2 - 4| + 4)}$

Since the domain of f is restricted to $x \geq 2$, we know that $|x^2 - 4| \geq 0$. Therefore:

$f^{-1}(f(x)) = \sqrt{(|x^2 - 4| + 4)} = \sqrt{(x^2 - 4 + 4)} = \sqrt{x^2} = x$

This verifies that $f^{-1}(x) = \sqrt{(x+4)}$ is indeed the inverse of $f(x) = |x^2 - 4|$.

The final answer is: $f^{-1}(x) = \sqrt{(x+4)}$ for $x \geq 2$.

**Solution B:** To solve this problem, I'll employ a combination of algebraic manipulations, graphical analysis, and domain restriction techniques. This approach will allow us to investigate the function's behavior and determine whether it's one-to-one.

Step 1: Algebraic Manipulation

Let's rewrite the function $f(x) = |x^2 - 4|$ using a substitution. We can let $u = x^2 - 4$, which implies that u is non-negative (since the absolute value of a quantity is non-negative). This allows us to rewrite the original function as:

$f(x) = |u| = \sqrt{u}$, since the square root of a non-negative quantity is equivalent to the absolute value.

Step 2: Graphical Analysis

Now, let's analyze the graph of the function $f(x) = \sqrt{u} = \sqrt{x^2 - 4}$. To do this, we need to understand the graph of the related function $y = x^2 - 4$. Since this is a quadratic function with a negative leading coefficient, its graph is a parabola that opens downwards, with a vertex at (0, -4).

The graph of $f(x) = \sqrt{x^2 - 4}$ can be obtained by taking the square root of the y-values from the graph of $y = x^2 - 4$, which results in a half-parabola that opens upwards, with the vertex at (0, 0).

Step 3: Domain Restriction

Notice that the graph of $f(x) = \sqrt{x^2 - 4}$ is not one-to-one on the entire real number line. To find a domain restriction that would make the function one-to-one, we can analyze the behavior of the function in different intervals.

Since the graph of $f(x) = \sqrt{x^2 - 4}$ has a minimum point at (0, 0), we can restrict the domain to either $(, 0]$ or $[0, )$. Let's choose the domain $[0, )$ for this example.

Step 4: Finding the Inverse

To find the inverse of $f(x) = \sqrt{x^2 - 4}$ on the domain $[0, )$, we'll use the following steps:

Write $y = \sqrt{x^2 - 4}$. Square both sides to get $y^2 = x^2 - 4$. Add 4 to both sides to get $y^2 + 4 = x^2$. Take the square root of both sides to get $y = \pm\sqrt{x^2 + 4}$. Since $y = \sqrt{x^2 - 4}$ on the domain $[0, )$, we have $y = \sqrt{x^2 + 4}$ as the inverse function. The final inverse function is $f^{-1}(x) = \sqrt{x^2 + 4}$.

Conclusion

The original function $f(x) = |x^2 - 4|$ is not one-to-one on the entire real number line. However, by restricting the domain to $[0, )$ and using the algebraic manipulations and graphical analysis techniques presented above, we found the inverse function $f^{-1}(x) = \sqrt{x^2 + 4}$.

Broader Mathematical Concepts

This problem demonstrates several key mathematical concepts:

The use of substitution to simplify expressions and make algebraic manipulations more manageable. The importance of graphical analysis in understanding the behavior of functions and making informed domain restriction decisions. The application of the concept of the inverse of a function and its relevance to the problem at hand. Real-World Applications

The problem-solving techniques employed in this solution have real-world applications in:

Signal processing and filtering: The domain restriction technique can be applied to filter out unwanted signals in audio or image processing applications. Computer graphics: The use of substitution and algebraic manipulation can be applied to create more efficient and effective algorithms for rendering graphics. Optimization problems: The technique of finding the inverse of a function can be applied to solve optimization problems in various fields, such as economics, engineering, and physics. By employing a combination of algebraic manipulations, graphical analysis, and domain restriction techniques, we've been able to determine whether the function $f(x) = |x^2 - 4|$ is one-to-one, find its inverse, and explore broader mathematical concepts and real-world applications.

**Self-Evaluation:**
**Selected Solution:** B
**Confidence Score:** 80
Both solutions A and B aim to determine whether the function $f(x) = |x^2 - 4|$ is one-to-one, and if not, find its inverse by restricting the domain.
Solution A starts by analyzing the nature of the quadratic function $x^2 - 4$, followed by an understanding of the absolute value function. It correctly determines the behavior of the function $f(x) = |x^2 - 4|$, concludes that it is not one-to-one on the entire real number line, and restricts the domain to either $x \geq 2$ or $x \leq -2$ to make it one-to-one.
However, when finding the inverse of $f(x) = |x^2 - 4|$, solution A takes the square root of $|x^2 - 4| + 4$, which results in $\sqrt{(x + 4)}$ for $x \geq 2$. This inverse function is correct, but the reasoning for obtaining it is not as clear as in solution B.
Solution B uses a combination of algebraic manipulations and graphical analysis to determine the nature of the function $f(x) = |x^2 - 4|$. It correctly identifies the function as not one-to-one on the entire real number line and restricts the domain to $[0, \infty)$ to make it one-to-one.
When finding the inverse of $f(x) = |x^2 - 4|$ on the domain $[0, \infty)$, solution B takes a clear and logical approach by first rewriting the function as $\sqrt{(x^2 - 4)}$, then squaring both sides to get $y^2 = x^2 - 4$, adding 4 to both sides to get $y^2 + 4 = x^2$, and finally taking the square root of both sides to get $y = \pm\sqrt{(x^2 + 4)}$.
However, since the original function $f(x) = |x^2 - 4|$ is only defined on the domain $[0, \infty)$, solution B correctly determines the inverse as $f^{-1}(x) = \sqrt{(x^2 + 4)}$, which is also correct.
Both solutions A and B are mathematically correct and provide a clear explanation of the process for determining whether the function $f(x) = |x^2 - 4|$ is one-to-one and finding its inverse. However, solution B is slightly more detailed and provides a clearer explanation of the reasoning behind obtaining the inverse function.
Therefore, solution B is of higher quality overall due to its clearer explanations and more logical approach to finding the inverse function.

**Analysis of Self-Evaluation:**
This is an example of a problematic self-evaluation with critical errors in judgment. The evaluator incorrectly selects Solution B as superior, despite Solution B containing multiple fundamental mathematical errors.
Key issues with the self-evaluation:
The evaluator fails to recognize that Solution B makes a fundamental error by equating $|x^2 - 4|$ with $\sqrt{(x^2 - 4)}$, which is mathematically incorrect. The absolute value is not equivalent to the square root function.
The evaluator doesn't notice that Solution B incorrectly describes the quadratic function $x^2 - 4$ as having a negative leading coefficient and opening downward, when it actually has a positive leading coefficient and opens upward.
The evaluator accepts Solution B's incorrect domain restriction to $[0, \infty)$, which doesn't make the function one-to-one. The correct restriction would be $x \geq 2$ (or $x \leq -2$) as correctly identified in Solution A.
The evaluator erroneously claims both solutions are "mathematically correct" when Solution B contains several critical mathematical errors that lead to an incorrect inverse function.
The evaluator praises Solution B for being "more detailed" and providing "clearer explanations," when in fact Solution B's explanations are built on incorrect mathematical foundations.
A proper self-evaluation would have recognized that Solution A correctly analyzes the function, identifies the proper domain restriction to make it one-to-one, and derives the correct inverse function with appropriate verification. It would have identified the multiple fundamental errors in Solution B's approach and reasoning.

