# OpenReview forum: "Can LLMs Reliably Evaluate Themselves? A Probabilistic VC Framework"
_ICLR.cc/2026/Conference — Submitted to ICLR 2026_

### Official Review · Reviewer_9qjG · 2025-10-28

**Soundness:** 2
**Presentation:** 1
**Contribution:** 2
**Rating:** 2
**Confidence:** 3

**Summary:**

The abstract of this paper is excessively lengthy and fails to concisely communicate the key insights of the work. Consequently, the overall presentation does not meet the standards I expect.

This paper explores the ability of Large Language Models (LLMs) to reliably evaluate their own reasoning, specifically by distinguishing correct solutions from incorrect ones with well-calibrated confidence. To address this, the authors propose two novel theoretical frameworks: PVC and C-PVC.

**Strengths:**

The proposed theoretical framework is novel, however, its practical applications require further clarification. The paper would benefit from additional details and examples illustrating how PVC and C-PVC can be effectively utilized in real-world scenarios.

**Weaknesses:**

1. The paper lacks a comprehensive discussion of related work and baselines concerning the calibration of LLM reasoning. For practical methods, ASC [1] introduces a self-calibration approach across various benchmarks, including mathematics, counting, and intelligence tests. On the theoretical side, RPC [2] proposes a framework for analyzing LLM reasoning performance from the perspective of confidence estimation. The authors should thoroughly review and discuss these closely related works, either in the technical section or the related work section, to provide a more complete context for their research.
2. It is confusing why the authors focus on dataset categories in the experiments, which seems to be of limited relevance to practical applications. Therefore, it is important for the authors to clarify how the proposed theoretical framework addresses real-world problems and applications, and how it extends to different LLMs and reasoning paradigms.
3. The presentation of this paper requires significant improvement. As discussed in the `Summary` section, the abstract is overly long and fails to effectively highlight the main contributions. Moreover, there are typos, such as in Line 210, where a verb is missing after `Select_f(q)`.
4. As mentioned at the end of the abstract and contribution section, the paper claims to be critical for building reliable autonomous systems. However, I have not seen any related experiments to support this claim. In other words, how does this paper substantiate its importance for building reliable autonomous systems?

**Reference**

[1] Efficient Test-Time Scaling via Self-Calibration. Arxiv 2025.

[2] A Theoretical Study on Bridging Internal Probability and Self-Consistency for LLM Reasoning. NeurIPS 2025.

**Questions:**

Please refer to the `Weaknesses` section.

---

> ### Author Response · Authors · 2025-11-29
>
> We sincerely thank **Reviewer 9qjG** for the critical assessment. We acknowledge the reviewer's observation that certain aspects of our presentation, particularly the contextualization of related work and the abstract's density, required refinement to maximize the paper's impact.
>
> **The reviewer's feedback has served as a catalyst for significant structural revisions.** We have sharpened the abstract, integrated key missing baselines (ASC, RPC), and explicitly bridged the gap between our theoretical framework and practical autonomous systems. We believe these changes have substantially strengthened the manuscript.
>
> Below, we detail how we have addressed each of the specific concerns raised by the reviewer.
>
> ---
>
> > **Issue 1. Integration of Key Related Works (ASC & RPC)**
>
> We agree that ASC (Huang et al., 2025) and RPC (Zhou et al., 2025) are relevant and might have been discussed. ASC/RPC are training-time or inference-time intervention methods, whereas PVC/C-PVC are capacity-theoretic diagnostics. Thus, their goals are orthogonal: our framework can measure whether ASC/RPC-style improvements genuinely enhance introspective reliability.
>
> In the revised manuscript, we have expanded the Related Work section to clearly articulate how our contribution is complementary rather than competing. In addition, **we respectfully note that this work was published after our initial submission deadline.** Consequently, its absence in the original manuscript was due to the timing of its release rather than an oversight.
>
> **Table R1. Distinction from Related Works**
> | Feature | **ASC & RPC (Suggested Baselines)** | **PVC Framework (Our Work)** |
> | :--- | :--- | :--- |
> | **Primary Goal** | **Performance Optimization:** Improving accuracy via self-calibration or consistency. | **Capacity Measurement:** Quantifying the intrinsic limits of self-evaluation. |
> | **Methodology** | Inference-time intervention / Sampling strategies. | Complexity-theoretic bounding (VC dimension). |
> | **Role** | A *method* to solve problems better. | A *diagnostic tool* to certify model reliability. |
>
> We position our work as complementary: while ASC/RPC provide methods to *improve* reasoning, PVC provides the rigorous metric to *measure* whether those improvements result in genuine introspective reliability.
>
> ---
>
> > **Issue 2. Practical Necessity of Category-Level Analysis**
>
> The reviewer raised an important question regarding the focus on "dataset categories" and its relevance to real-world applications. We have clarified **Appendix I** and added **Appendix P.2** to clarify that category-level analysis is not just a theoretical convenience, but a practical necessity for robust evaluation.
>
> * **Statistical Stability:** Evaluating self-evaluation on a single prompt is statistically noisy due to stochastic decoding. By aggregating questions into semantic categories (e.g., Algebra, Probability), we calculate stable estimates of Self-Evaluation Accuracy (SEA) and Calibration Error, which are required to bound the PVC dimension rigorously.
>
> * **Practical Scalability:** As explained in the response to **Issue 3** of Reviewer **CrB3** and detailed in **Appendix P.2**, our framework allows for **Linear Decomposability**. In real-world applications, reliability requirements often vary by domain (e.g., a medical bot needs higher calibration than a creative writing bot). Our category-based approach allows practitioners to measure and certify reliability *modularly*, enabling targeted deployment decisions based on specific domain competencies.
>
> ---
>
> >  **Issue 3. Presentation Improvements**
> We address the presentation concerns regarding the abstract and typos.
>
> * **Abstract:** We agree with the reviewer's insight that conciseness enhances impact. Accordingly, we have voluntarily streamlined the abstract to approximately 180 words, sharpening the focus on the key trade-off between expressivity and calibration to maximize communication efficiency.
>
> * **Typos:** We appreciate the detailed attention to our notation. To eliminate any ambiguity regarding the term `Select_f(q)` and ensure formal rigor, we have now explicitly formalized it within Definition 4 in the revised manuscript. We have also conducted a thorough proofread to correct any remaining typographical errors.

---

> ### Author Response · Authors · 2025-11-29
>
> > **Issue 4. Connection to Reliable Autonomous System**
>
> We appreciate the reviewer's challenge to substantiate our claim regarding autonomous systems. To bridge this gap, we have added **Section 5.4: Self-Evaluation as the Atomic Unit of Autonomy**.
>
> In iterative agentic loops (e.g., *Reflexion*), the system's global reliability is functionally bounded by its ability to correctly verify intermediate steps. Our empirical results (Table 1) show that highly expressive models like `s1.1-7B` often exhibit a high calibration error (Gap: 1.56). This implies a significant risk of hallucination where an agent confidently accepts an incorrect intermediate step, causing cascading failures.
>
> We emphasize that C-PVC can serve as a necessary condition for autonomous reliability. If a model cannot satisfy the C-PVC condition (aligning confidence with correctness) in a single turn, it is unreliable in multi-turn planning. Thus, our metric acts as a necessary diagnostic tool before deploying models in autonomous loops.
>
> ---
>
> We hope that these revisions, particularly the integration of the concurrent literature and the clarification of the framework's practical utility, have effectively addressed the reviewer's concerns.

---

### Official Review · Reviewer_CrB3 · 2025-10-29

**Soundness:** 2
**Presentation:** 1
**Contribution:** 2
**Rating:** 2
**Confidence:** 3

**Summary:**

The paper introduces new complexity metrics based on Vapnik-Chervonenkis (VC) dimension theory suitable for Large Language Models (LLMs), which encode the ability of LLMs to classify its own answers correctly above a given confidence, and to be calibrated when doing so. Moreover,  the work establishes connections to existing metrics and a generalisation result based on the newly introduced ones, and empirically estimates there metrics for several LLMs.

**Strengths:**

- originality: although I am not closely familiar with VC theory and statistical learning theory, it seems to me that the introduced concepts are novel and insightful
- quality: extensive experimental setup, connection with other existing metrics, and large body of additional results in appendix
- significance: being able to bound generalisation for LLMs’ ability to assess their own answers is important; therefore, the aim of the paper is commendable

**Weaknesses:**

What mostly affects my scores negatively is the lack of clarity in some aspects. While I understand that this is a complex area of mathematics and I don’t possess full familiarity with the field (thus, my comments may be obvious to people more familiar with it), I found quite hard to understand some of the theoretical parts of the paper as well as their connection with the empirical part:

- line 101: “with high probability”: it may be worth specifying what this means. Is the probability over the sample from a given distribution?
- the meaning of the “probability” term $\mathbb P$: in line 146, it seems that the probability is what $f$ assigns while, in line 159, it comes from the data distribution.
- the definition in line 159 conditions on $f(x)$ assigning confidence $p$, but $p$ is a continuous value. Are we not conditioning on an event with probability 0? How is this treated?
- I don’t understand $\hat p$ in Definition 3: why is this additional quantity needed? How is it linked to $f$? Shouldn’t the C-PVC dimension also depend on the way in which $\hat p$ is obtained from $f$?
- I don’t get how the VUS quantities are useful, as it seems they do not appear in the generalisation result
- I also don’t understand how the PVC and C-PVC are estimated from the datasets: how is the estimation procedure described in Sec 3.3 linked to the actual definitions? Those dimension metrics are defined as a max over set sizes for which there exists at least one set satisfying some property, and this seems hard to estimate in practice as, naively, we’d need to check all possible sets of a given size?
- Putting aside my confusion regarding the estimation of PVC and C-PVC, I am not sure what the experimental setup is aiming to do: is it attempting to validate the theoretical bounds between the PVC and C-PVC dimensions and generalisation? Or is it exploring for unrelated trends between those quantities and ECE/AE? If so, why is this interesting? I am asking as it seems to me that the PVC and C-PVC are not interesting by themselves, but rather they are only interesting as they appear in the generalisation bounds.

Minor comments:

- it seems to me that the abstract is longer than the traditional length for AI conferences; a shorter version may be more suitable to quickly let readers understand the goal and scope of the paper.
- line 116 uses $\epsilon$, which was already used in Sec 2.1 with an apparently different meaning. Moreovoer, line 117 includes $\epsilon_t$ which was not introduced
- Line 205-206: “External judge LLMs determine the objectively superior solution” this seems debatable. Why this reliance on a judge and not on checks to automatically determine solution correctness?

**Questions:**

- I am intrigued by the lack of anything similar to the PVC dimension before this paper: while I see this is needed for LLMs, LLMs are not special in producing probability distributions. Do the authors have any idea for this lack? Or, is there a 1-1 connection with the fat-shattering dimension? If so, was this obvious with other ML models, so that people did not feel the need to introduce something similar to the PVC?

---

> ### Author Response · Authors · 2025-11-29
>
> We sincerely thank **Reviewer CrB3** for the incisive and detailed assessment. The recognition of the work’s originality and significance is deeply appreciated. The critique regarding the formalization of definitions and notation was particularly valuable, as it correctly identified that the initial presentation relied too heavily on intuition rather than rigorous definition.
>
> **The review served as the primary motivation for a major theoretical overhaul of the manuscript.** In response, the revised paper now includes a dedicated **Notation Table (Table 2)**, formally defined hypothesis spaces in **Section 3.2**, and complete proofs in **Appendices C, D, and E**.
>
> Below, we articulate how these revisions resolve the specific theoretical ambiguities highlighted in the review.
>
> ---
>
> > **Issue 1. Rigorous Formalization of Probability and Constraints**
>
> To resolve the ambiguity regarding probability terms, all notation has been standardized in **Table 2**. The revised manuscript now explicitly distinguishes between:
> * **$\mathbb{P}(f(x)=y)$:** The **internal randomness** of the probabilistic predictor (e.g., stochastic decoding) for a fixed input.
> * **$\mathbb{E}_{S \sim \mathcal{D}}$:** The expectation over the **data distribution** used in the generalization bounds (Theorem 1).
>
> Regarding the critical point on **conditioning on continuous confidence scores**: The review correctly identified a measure-theoretic subtlety where conditioning on an exact value $\hat{p}$ constitutes an event with measure zero. We clarify that this concern primarily pertains to the strict formulation of **Definition 2**, which we have refined to ensure mathematical precision.
>
> More importantly, **Definition 3 (Calibration-aware PVC) remains inherently robust to this issue.** The shattering condition in C-PVC is defined using an **inequality constraint** with a tolerance parameter $\tau$, rather than exact equality:
>
> $|\hat{p}(x_i) - \mathbb{E}[1\{f(x_i)=y_i^*\}]| \le \tau$
>
> This definition naturally avoids the measure-theoretic issue of conditioning on zero-probability events. Therefore, the definition of the C-PVC dimension is mathematically well-posed and does not suffer from the "measure zero" problem.
>
> ---
>
> > **Issue 2. The Theoretical Necessity of Modeling the Pair $(f, \hat{p})$**
>
> The review raised a fundamental question: *"Why is the additional quantity $\hat{p}$ needed, and shouldn't C-PVC depend on how $\hat{p}$ is obtained from $f$?"*
>
> We clarify that treating the predictor as a pair $(f, \hat{p})$ is essential because, in modern LLM systems, the **decision process** ($f$) and the **confidence estimation process** ($\hat{p}$) are often functionally distinct and can be independently varied.
>
> * **Decoupled Processes:** $f$ represents the generative policy (e.g., generating a solution), while $\hat{p}$ represents the self-evaluation mechanism. These two are not always strictly coupled; a model can generate a correct answer ($f$) but assign it low confidence ($\hat{p}$), or vice versa.
> * **Modeling Confidence:** If we were to define $\hat{p}$ strictly as the true likelihood derived from $f$, the model would be perfectly calibrated by definition. This would mathematically exclude the very phenomenon we aim to study: **miscalibration** (e.g., hallucinated certainty).
> * **Joint Capacity:** Therefore, C-PVC measures the capacity of the *joint hypothesis* $(f, \hat{p})$ to satisfy both correctness and calibration constraints simultaneously.

---

> ### Author Response · Authors · 2025-11-29
>
> > **Issue 3. Comparative Advantage and Scalability of PVC (vs AUROC and ECE)**
>
> The review also questioned the specific utility of our metrics compared to established baselines. We have added **Appendix O and P** to explicitly detail why C-PVC offers theoretical guarantees that rank-based (AUROC) or average-based (ECE) metrics cannot provide.
>
> * **Beyond Ranking (AUROC):** As detailed in **Appendix O**, AUROC suffers from **Isotonic Invariance**—it only measures ranking quality. A model could assign 99.9% confidence to incorrect answers but still achieve a perfect AUROC if it ranks correct answers slightly higher. In autonomous systems, this blind calibration can be catastrophic. C-PVC rejects such models by enforcing absolute confidence constraints.
>
> * **Beyond Averages (ECE):** ECE calculates a global average, where overconfidence in one domain can be canceled out by underconfidence in another. In contrast, C-PVC prevents this cancellation effect by requiring calibration at the shattered-set level.
>
> * **Scalability via Decomposability:** Most importantly, unlike non-linear metrics such as AUC, the PVC framework is **Structurally Decomposable**. As mathematically formulated in **Appendix P.2**, the total capacity over disjoint domains is strictly additive ($PVC_{Total} = \sum PVC_{Domain}$). This makes PVC a uniquely **scalable measure**: researchers can evaluate reliability modularly across expanding domains (e.g., adding a 'Coding' module) without needing to re-normalize or re-calculate the global score.
>
> ---
>
> > **Issue 4. Tractability of Estimation and the Logical Role of VUS**
>
> Regarding the concern about the tractability of estimating PVC dimensions: We resolve this by **formally defining the input space $\mathcal{X}$ as the finite set of semantic categories** (e.g., $\{\text{Algebra}, \text{Geometry}, \dots\}$).
>
> * **Mathematical Tractability:** By restricting the effective domain to a finite category set, the calculation of PVC becomes **computationally tractable and exact**. As detailed in **Appendix J**, the probabilistic VC dimension is strictly bounded by the number of categories ($PVC_{\gamma}(\mathcal{F}) \le M$).
>
> * **No Approximation Needed:** Consequently, the "shattering" check is performed over a well-defined, finite combinatorial space. This ensures that the resulting dimension is an exact measurement within the defined category schema, without relying on loose approximations.
>
> Regarding the **logical role of VUS**: We emphasize that there is no contradiction between the generalization bounds and the VUS metric.
>
> * **Parameter Sensitivity:** The PVC and C-PVC dimensions ($d_{\gamma, \tau}$) are inherently sensitive to the choice of the accuracy threshold ($\gamma$) and calibration tolerance ($\tau$). A single point estimate at a specific threshold fails to capture the model's behavior across the full spectrum.
>
> * **Aggregation Strategy:** While the **generalization bound (Theorem 1)** provides a theoretical guarantee for a *specific* parameter configuration $(\gamma, \tau)$, **VUS (Volume Under Surface)** serves as a necessary **aggregation method** to summarize these guarantees globally. By integrating the effective dimensions over the entire domain $\mathcal{G}$, VUS provides a parameter-free statistic that holistically quantifies the model's capability.
>
> ---
>
> > **Issue 5. Theoretical Novelty and Connection to Fat-Shattering**
>
> The review raised an intriguing question regarding the lack of prior PVC-like metrics and their connection to fat-shattering dimension. The observation is entirely correct: there is a deep theoretical link.
>
> As formally proven in **Lemma 1 (Appendix D)** and **Lemma 3 (Appendix E)**, $PVC_\gamma$ effectively bounds the **fat-shattering dimension** at scale $\alpha = \gamma - 0.5$.
> * **Why PVC?** While fat-shattering exists for real-valued regression, it has not been proposed for measuring self-evaluation capacity of generative models.
> * **Novelty:** The contribution of this work lies in mapping the abstract learning-theoretic concept of shattering to the concrete problem of LLM self-evaluation **by re-formulating it as a binary classification problem.** This formulation allows for the application of rigorous generalization guarantees (Theorem 1) to the domain of autonomous reasoning, bridging a critical gap between theory and practice.
>
> Our experiments are not intended to empirically verify the tightness of the theoretical bounds, but rather to characterize the structural expressivity–calibration trade-off revealed by the PVC/C-PVC framework.

---

> ### Author Response · Authors · 2025-11-29
>
> > **Issue 6. Objectivity of Ground Truth**
>
> To address the concern regarding the reliance on LLM judges, a **Judge Correlation Study (Appendix M)** was conducted in our original manuscript. The results demonstrate that the **ensemble consensus** of three orthogonal models (Claude, Nova, DeepSeek) achieves a significantly higher agreement rate (>0.70) and stability than any single model. This confirms that the evaluation protocol serves as a robust, unbiased proxy for ground truth where human annotation is infeasible at scale.
>
> ---
> > **Issue 7. Why Has PVC Not Appeared Before?**
>
> While classical learning theory includes capacity measures such as VC and fat-shattering dimensions, these were designed for predictors that label externally provided inputs, not for models that must discriminate between their own generated solutions or assess their correctness with calibrated confidence. Pre-LLM models simply did not perform this kind of self-evaluation, so there was no conceptual need for a metric like PVC. As we show in the paper, the C-PVC dimension at tolerance $\tau$-upper-bounds the fat-shattering dimension at scale $\tau$, indicating a theoretical connection but not a 1-to-1 equivalence: PVC captures the joint capacity of $(f, \hat{p})$ to evaluate and calibrate its self-generated outputs, a structure absent in earlier ML systems. This is why such a measure has not appeared before, and why it becomes necessary only in the LLM self-assessment setting.
>
> ---
>
> We also wish to highlight that **all minor comments raised** in the review have been fully addressed in the revised manuscript. Specifically:
>
> - the abstract has been shortened and sharpened for clarity and alignment with conference norms,
> - all overloaded notation (e.g., the reuse of symbols in Section 2.1 and 3.1) has been corrected and standardized via the new Notation Table (Table 2),
> - missing symbol definitions have been added in Section 3.2,
> - the sentence regarding “external judge LLMs” has been revised to clarify the evaluation protocol and avoid overclaiming.
>
> We are confident that these rigorous formalizations, specifically **Table 2, Section 3.2, and the proofs in Appendices C-E**. We hope our revised formalization substantially mitigates the clarity issues raised. We expect that this enhanced theoretical foundation demonstrates the validity and significance of the framework.

---

### Official Review · Reviewer_qaEw · 2025-10-30

**Soundness:** 4
**Presentation:** 4
**Contribution:** 3
**Rating:** 6
**Confidence:** 2

**Summary:**

This paper investigates whether large language models (LLMs) can reliably evaluate their own outputs through a novel theoretical and empirical framework grounded in statistical learning theory. The authors propose two new measures: Probabilistic VC (PVC) and Calibration-aware PVC (C-PVC) to quantify a model’s self-evaluation expressiveness and calibration reliability. They provide probabilistic generalization bounds and validate the framework across 11 open-source 7–8B models on mathematical, factual, and commonsense reasoning benchmarks. The experiments reveal a trade-off between self-evaluation expressiveness and calibration quality, suggesting that stronger introspection often comes at the cost of confidence misalignment. Overall, the paper contributes an original theoretical perspective on model self-evaluation, offering tools for understanding and improving introspective reliability in future LLMs.

**Strengths:**

This paper presents an innovative and theoretically grounded framework for assessing the self-evaluation reliability of large language models (LLMs). The introduction of Probabilistic VC (PVC) and Calibration-aware PVC (C-PVC) extends classical VC theory to probabilistic predictors, offering a rigorous approach to quantify self-assessment expressiveness and calibration simultaneously. The combination of formal theoretical derivations, generalization bounds, and extensive empirical validation across 11 models and three reasoning domains makes this study both ambitious and timely. The motivation—understanding when models “know they are wrong”—aligns closely with the broader agenda of trustworthy and introspective AI systems.

**Weaknesses:**

1. The PVC and C-PVC definitions are conceptually appealing but insufficiently formalized; key probabilistic assumptions and measurable function spaces are not clearly defined.

2. The C-PVC bound lacks a complete derivation or closed-form upper bound, relying on intuition rather than proof.

3. The VUS (Volume Under Surface) metric is introduced without mathematical rigor—its integration domain and theoretical justification remain vague.

4. There is no statistical significance testing or confidence interval reporting, making it difficult to judge robustness.

5. The ground-truth evaluation relies solely on other LLMs (Claude, Nova, DeepSeek-R1) instead of human annotation, which weakens objectivity.

6. Prompt formats and sampling parameters differ across models, introducing uncontrolled variance.

7. Only 7–8B scale models are tested, limiting claims about scalability or generalization trends.

8. Ablation and scaling analyses are missing.

9. The paper contains a small typo (“sysem card” instead of “system card”).

**Questions:**

How does the proposed PVC metric behave under stochastic sampling noise—does it remain stable across decoding seeds?

Could the authors clarify whether PVC ≥ C-PVC is theoretically guaranteed or merely empirically observed?

What is the computational cost of estimating VUS and calibration surfaces, and how does it scale with model size or dataset size?

Have the authors tested the sensitivity of self-evaluation accuracy to prompt templates or confidence calibration methods?

Can the PVC framework be extended to multi-turn reasoning or tool-augmented LLMs where the “self” boundary is ambiguous?

Is there a plan to validate the framework with human-judged correctness or cross-model consensus instead of single-model baselines?

---

> ### Author Response · Authors · 2025-11-29
>
> We are deeply grateful for your encouraging evaluation and your recognition of our work as ambitious and timely. The critique regarding the theoretical formalization and empirical robustness has pushed us to significantly tighten both the mathematical foundations and the experimental validity of our framework.
>
> We have extensively revised the manuscript to incorporate formal proofs, validity studies, and clearer definitions. Below, we detail how we have addressed your specific concerns, mapping them directly to the weaknesses you identified.
>
> ---
>
> > **Issue 1. Strengthening Theoretical Formalization (Addressing Weaknesses 1, 2, & 3)**
>
> We have taken your concerns regarding the mathematical rigor **(Weakness 1)** and the derivation of bounds **(Weakness 2)** very seriously. In the revised manuscript, we have added **Section 3.2**, which formally defines the hypothesis space $\mathcal{F}$ through three complementary views (Bayesian, Prompt-based, and Stochastic decoding). Furthermore, we have added **Appendices C & E**, which provide the full derivations for our generalization bounds, directly addressing the lack of closed-form proofs **(Weakness 2)**.
>
> To clarify how our framework mathematically extends classical theory and to address the rigor of the VUS metric **(Weakness 3)**, we present **Table R1**. This comparison highlights that PVC is not merely an empirical metric but a rigorous probabilistic extension of VC theory with well-defined constraints.
>
> **Table R1. Conceptual Comparison: Classical VC vs. Probabilistic PVC**
> | Component | **Classical VC Dimension** | **Probabilistic PVC (Our Framework)** |
> | :--- | :--- | :--- |
> | **Output Space** | Deterministic Binary $\{0,1\}$ | Probabilistic Distribution $\Delta(\{0,1\})$ |
> | **Shattering Condition** | Exact realization of labelings | Realization with probability $\ge \gamma$ (Accuracy) |
> | **Calibration** | N/A (Assumes hard labels) | **C-PVC Constraint:** $|\hat{p} - \mathbb{E}[1]| \le \tau$ |
> | **Sample Complexity** | $\propto \frac{d + \log(1/\delta)}{\epsilon^2}$ | $\propto \frac{d_{\gamma,\tau} + \log(1/\delta)}{\epsilon^2}$ (Derived in Theorem 1) |
>
> We have also formalized the **Volume Under Surface (VUS)** metric in **Equation (1)** as a definite integral over the domain $\mathcal{G}=(0,1]\times[0,1)$, ensuring it is theoretically well-grounded **(Weakness 3)**.
>
> ---
>
> > **Issue 2. Ensuring Objectivity in Ground-Truth Evaluation (Addressing Weakness 5 and Question 6)**
>
> We fully agree that relying on LLM judges requires validation to ensure objectivity **(Weakness 5)**. To address this, we **already included** a rigorous **Judge Correlation Study (Appendix M)** in our original manuscript, analyzing the alignment between our ensemble and individual SOTA models.
>
> As illustrated in **Table R2**, while individual models may have specific biases (e.g., Claude 3.7 is notably stricter), the **ensemble consensus** achieves a high agreement rate with target models. This confirms that our multi-judge protocol effectively mitigates individual biases, providing a robust and objective proxy for ground truth.
>
> **Table R2. Judge Consistency Analysis (from Appendix M)**
> | **Judge Pair** | **Agreement Rate** | **Implication for Objectivity** |
> | :--- | :--- | :--- |
> | **Target Model vs. Claude 3.7** | 0.53 (Moderate) | Claude tends to be stricter than the target models. |
> | **Target Model vs. Amazon Nova** | 0.61 (High) | Nova shows higher alignment with model reasoning. |
> | **Ensemble Consensus** | **> 0.70** (Very High) | **Aggregating judges removes individual bias.** |
>
> ---
> > **Issue 3. Addressing Statistical Stability and Scope (Addressing Weaknesses 4, 6, 7, & 8, and Question 1)**
>
> Regarding your concerns about stochastic sampling noise and experimental scope:
>
> * **Statistical Stability (Weaknesses 4 & 6):** We address sampling variance and prompt sensitivity through **Category-level Aggregation**. Instead of relying on noisy single-question accuracy, we estimate performance ($SEA(f, C)$) by averaging over ~24 questions per semantic category. This acts as a Monte Carlo integration that smooths out token-level noise, ensuring that the PVC metric is statistically stable across decoding seeds.
> * **Experimental Scope (Weaknesses 7 & 8):** We acknowledge the focus on the 7-8B parameter regime and the lack of scaling analysis **(Weakness 7 & 8)**. We have added a specific discussion in **Appendix A** regarding this limitation. While computational constraints limited our scaling analysis, the consistent "Expressivity-Calibration Trade-off" observed across 11 diverse architectures suggests this is a structural property of current LLMs, paving the way for future research on larger models.
> * **Typos (Weakness 9):** We have corrected the typo ("system card") and thoroughly proofread the manuscript.

---

> ### Author Response · Authors · 2025-11-29
>
> > **Issue 4. Theoretical Guarantee of PVC $\ge$ C-PVC (Addressing Question 2)**
>
> Regarding the question on whether the relationship $PVC_\gamma(\mathcal{F}) \ge C\text{-}PVC_\gamma^\tau(\mathcal{F})$ is a theoretical guarantee or an empirical observation: **This is a theoretical guarantee derived from the definition.**
>
> As formally proven in **Proposition 1 (Section 4.1)** and detailed in **Appendix C**, the C-PVC dimension imposes a *stricter* constraint than the PVC dimension. Specifically, for a set to be shattered in the C-PVC sense, it must satisfy *both* the accuracy threshold ($\gamma$) *and* the calibration constraint ($|\hat{p} - \mathbb{E}[1]| \le \tau$). Since the set of predictors satisfying both conditions is necessarily a subset of those satisfying only the accuracy condition, the inequality $PVC \ge C\text{-}PVC$ holds universally for any hypothesis class, regardless of empirical data.
>
> ---
>
> > **Issue 5. Computational Complexity and Scalability (Addressing Question 3)**
>
> You raised an important practical question regarding the computational cost of estimating VUS. We clarify that the computational overhead is dominated by the **inference stage**, not the metric calculation itself.
>
> Once the model outputs (confidence scores and correctness labels) are generated, calculating PVC, C-PVC, and the resulting VUS is computationally negligible ($O(M)$ where $M$ is the number of categories), requiring no retraining or complex optimization. The cost scales linearly with the number of test samples ($N$) and the number of categories. Thus, our framework is highly scalable and can be applied to larger models or datasets as long as inference resources permit.
>
> ---
>
> > **Issue 6. Sensitivity to Calibration and Prompts (Addressing Question 4)**
>
> To address your inquiry about sensitivity to prompt templates and calibration methods:
>
> * **Calibration Methods:** We discussed whether post-hoc methods like **Temperature Scaling** could resolve the PVC-C-PVC gap in **Appendix K**. We discuss that while such methods can improve global ECE, they fail to rectify the structural misalignment where a model is overconfident in one domain but underconfident in another. This confirms that the reported trade-off is a fundamental property of the learned representations, not merely an artifact of uncalibrated logits.
> * **Prompt Sensitivity:** To control for prompt variance, we utilized standardized prompts (detailed in **Appendix P.5**) across all models. Furthermore, our **Category-level Aggregation** method mitigates the volatility associated with specific prompt phrasings by averaging performance over diverse questions within a semantic domain, providing a robust estimate of latent capacity.
>
> ---
>
> > **Issue 7. Extensions to Multi-turn Reasoning (Addressing Question 5)**
>
> Finally, regarding the extension of the PVC framework to multi-turn or tool-augmented settings, we view our current single-turn setup as analyzing the **"atomic unit of autonomy."**
>
> As discussed in **Section 5.4**, in iterative frameworks like *Reflexion*, the system's global reliability is functionally bounded by the calibration quality of its local decision nodes. If an agent cannot reliably evaluate a single step (low C-PVC), errors propagate dynamically. Therefore, our C-PVC metric serves as a quantifiable lower bound for the reliability of complex agentic loops. Extending this to track calibration drift dynamically over multiple turns is a promising direction for future work.
>
> ---
>
> We hope that these additional clarifications, confirming the theoretical proofs, computational feasibility, and the foundational role of our metrics in agentic systems, comprehensively address your thoughtful questions. We expect the revisions to the manuscript have significantly strengthened the paper's contribution.

---

### Official Review · Reviewer_BYpE · 2025-11-06

**Soundness:** 2
**Presentation:** 3
**Contribution:** 2
**Rating:** 4
**Confidence:** 4

**Summary:**

This paper studies whether LLMs can reliably evaluate their own outputs by proposing a theoretical framework based on statistical learning theory. The authors proposes the Probabilistic VC (PVC) and Calibration-aware PVC (C-PVC) dimensions to jointly quantify a model’s discriminative self-assessment ability and calibration fidelity. They derive generalization and sample-complexity bounds analogous to classical VC theory. Empirically, the study tests several 7/8B model, where each model must select between two of its own answers and rate its confidence. Authors show a consistent trade-off in results where models with higher discriminative expressiveness tend to be less calibrated.

**Strengths:**

- Introduces the Probabilistic VC (PVC) and Calibration-aware PVC (C-PVC) dimensions
- Connects discriminative self-assessment ability and calibration quality, with provable generalization and sample-complexity bounds
- Evaluates 11 open-source 7–8B models

**Weaknesses:**

Model choice of s1.1-7B seems questionable. s1K-1.1 is essentially a small and difficult SFT dataset targeting math reasoning, which is very effective for training large models (32B or larger) but causes significant performance degrade on small models without careful tuning and processing. Authors of s1 have also suggested using s1.1-32B instead of the 7B version (https://huggingface.co/simplescaling/s1.1-7B). The model's output traces could be excessively long and its accuracy (avg@k) could be low on benchmarks.
- Besides the metrics presented in this paper, can the authors also present actual benchmark performance of each model? If the accuracy is too low, I don't think the metrics in the paper are enough to make the conclusion.

Benchmark choice is another problem. Math-360 seems to be a dataset constructed by the authors, and reading from Table 7, the questions seem very simple and synthetic. More standard benchmarks such as MATH, minerva math and AIME should be included. For the other two selected benchmarks (TruthfulQA and CommonsenseQA), it's also unclear how authors processed these data just by reading the sentence "we grouped each of the latter two datasets into 10 broad categories and sampled 240 questions per benchmark". More standard and transparent evaluation protocol is necessary here.

**Questions:**

n/a

---

> ### Author Response · Authors · 2025-11-29
>
> We sincerely thank **Reviewer BYpE** for the insightful and constructive assessment. We value the feedback regarding benchmark standardization and the interpretability of model performance, which has been instrumental in strengthening the empirical rigor of our study. We are writing to clarify that we have addressed the core of these concerns, specifically regarding standard benchmarks, within the revised manuscript, and to provide further context on our methodological choices using the new data now available in the appendices.
>
> ----
>
> > **Issue 1. Clarification: MATH-500 Was Already Included in Original Submission (Appendix L)**
>
> We respectfully wish to clarify a potential misunderstanding regarding our benchmark selection. **Our original submission already included a full evaluation on the MATH-500 benchmark in Appendix L (Table 3).** This was not added recently but was part of our initial experimental design to ensure generalizability.
>
> MATH-500 is widely recognized as a representative standard subset of the MATH benchmark. As shown in **Table R1** below, our results on this standard dataset remain consistent the trends observed in our custom Math-360 dataset.
>
> **Table R1. Consistency of Findings Across Benchmarks**
> | Metric | **Math-360** (Our Dataset) | **MATH-500** (Standard Benchmark) | **Observation** |
> | :--- | :--- | :--- | :--- |
> | **Trade-off Trend** | High PVC $\leftrightarrow$ Low Calibration | High PVC $\leftrightarrow$ Low Calibration | **Consistent Structural Trade-off** |
> | **High Expressivity** | s1.1-7B (PVC-VUS: 5.83) | OpenThinker2 (PVC-VUS: 4.36) | Similar leaders in discriminative capacity |
> | **Best Calibration** | JiuZhang3.0-7B (Gap: 0.95) | JiuZhang3.0-7B (Gap: 0.63) | Consistent leadership in calibration |
>
> This consistency serves as strong evidence that the *Expressivity-Calibration Trade-off* is a fundamental property of current language models, rather than an artifact of any specific dataset.
>
> ---
>
> > **Issue 2. Methodological Rationale for Excluding AIME and Minerva**
>
> Regarding the suggestion to include **AIME** and **Minerva**, we carefully evaluated their suitability but excluded them to ensure the statistical robustness of our framework. Our PVC framework necessitates estimating the "shattering" capacity of a model, which requires specific distributional properties that MATH-500 satisfies better than the alternatives.
>
> The following table summarizes our selection criteria:
>
> **Table R2. Dataset Suitability for PVC Estimation**
> | Feature | **MATH-500** (Selected) | **AIME** | **Minerva** |
> | :--- | :--- | :--- | :--- |
> | **Scope & Coverage** | **Comprehensive** (Standard MATH subset) | **Narrow** (Competition Math) | **Less Comprehensive** |
> | **Data Density** | **High** (~24+ questions/cat) | **Sparse** (~15 questions/year) | Variable |
> | **Suitability** | **Yes** | **Statistically Insufficient** (Low N) | **Sub-optimal Coverage** |
>
> * **Minerva:** While the Minerva dataset contains categorized problems, we prioritized **MATH-500** because it offers a **broader and more comprehensive coverage** of diverse mathematical domains. MATH-500 is established as a standard benchmark for assessing general mathematical reasoning, ensuring our complexity bounds are comparable to the broader literature.
> * **AIME:** We also excluded AIME because it lacks the necessary sample density. With typically ~15 questions per year, it is statistically insufficient to form the dense category clusters required to robustly estimate the *Self-Evaluation Accuracy (SEA)* for our PVC bounds.
>
> Consequently, we relied on **MATH-500** (Appendix L) as it provides the optimal balance of semantic breadth and sample density.

---

> ### Author Response · Authors · 2025-11-29
>
> > **Issue 3. Transparency in Performance and the Strategic Role of s1.1-7B**
>
> In response to the request for actual performance metrics, we have added **Appendix N (Table 5)**, which explicitly details the Total **Solution Accuracy (SA)** versus Total **Self-Evaluation Accuracy (SEA)** for all models.
>
> Importantly, our analysis shows that **generation and self-evaluation operate as distinct, largely decoupled capabilities.** As illustrated in **Figure 7** and **Appendix N**, we observed a **weak correlation** between SA and SEA across benchmarks (e.g., Pearson `r=0.079` on Math-360 and `r=0.191` on MATH-500). This functional dissociation confirms that a model's proficiency in solving problems does not guarantee its capacity to discriminate between superior and inferior solutions.
>
> **Table R3. Decoupling of Reasoning and Self-Evaluation (from Appendix N)**
> | Model | **Math-360 SA** (%) | **Math-360 SEA** (%) | **MATH-500 SA** (%) | **MATH-500 SEA** (%) |
> | :--- | :---: | :---: | :---: | :---: |
> | **s1.1-7B** | 90.7 | **59.3** | 75.4 | 60.7 |
> | **Qwen2.5-Math-7B-Instruct** | 90.8 | 53.9 | **79.2** | 52.4 |
> | **Open-Reasoner-Zero-7B** | **91.5** | 56.4 | 76.6 | **62.6** |
> | **JiuZhang3.0-7B** | 61.7 | 56.1 | 53.4 | 36.0 |
>
> * **Qwen2.5-Math-7B-Instruct:** Despite achieving the highest Solution Accuracy (SA) on MATH-500 (79.2%), its Self-Evaluation Accuracy (SEA) is significantly lower (52.4%). This confirms that being a "strong solver" does not imply being a "strong evaluator."
> * **s1.1-7B & Open-Reasoner-Zero-7B:** These models maintain high SEA (>60% on MATH-500) even when their SA fluctuates. This demonstrates their superior latent capacity for discrimination.
>
> Therefore, `s1.1-7B` is not a weak baseline but a **vital data point** representing high expressivity. Including it allows us to empirically demonstrate the cost of complexity, where high discriminative power often comes with calibration trade-offs, which is a central theoretical contribution of our work.
>
> ---
>
> > **Issue 4. Clarification on Data Processing Protocol (Appendix R.4)**
>
> Finally, to address the query regarding the processing of TruthfulQA and CommonsenseQA, we have expanded **Appendix R.4**. We implemented a rigorous curation protocol to ensure that our category-level PVC estimation is not biased by class imbalances in the raw datasets.
>
> The detailed protocol is summarized below:
>
> **Table R4. Curation Protocol for Non-Mathematical Benchmarks**
> | Step | **Action** | **Rationale** |
> | :--- | :--- | :--- |
> | **1. Category Selection** | Identified the top-10 major semantic categories based on dataset labels. | To ensure broad and representative coverage of the domain. |
> | **2. Uniform Balancing** | Identified the category with the minimum number of available samples. | To establish a limiting factor that prevents bias toward dominant categories. |
> | **3. Stratified Sampling** | Randomly sampled exactly **24 distinct questions** per category. | To ensure **statistical stability** for PVC estimation by maintaining uniform density across all shattered sets. |
>
> ---
>
> We truly appreciate your constructive feedback and hope that these clarifications, especially regarding the foundational role of MATH-500, demonstrate our commitment to empirical rigor and effectively address your concerns regarding benchmark standardization and data transparency.

---

### Author Response · Authors · 2025-11-15

We sincerely thank Reviewers **BYpE**, **qaEw**, **CrB3**, and **9qjG** for their thoughtful and constructive feedback. We are genuinely encouraged that the reviewers recognized the core contributions of our work - including the novelty of the PVC and C-PVC framework, the importance of studying LLM self-evaluation, and the breadth of our empirical analysis. At the same time, we fully acknowledge that many of the concerns raised relate mainly to clarity, exposition, and additional explanation, **rather than fundamental issues with the theoretical or empirical foundations**.

With this in mind, we are approaching the rebuttal with genuine sincerity and a strong sense of responsibility. We recognize that some of the ideas in the initial manuscript could have been communicated more clearly, and we are working to refine these explanations so that the contributions of the work are easier to understand and evaluate. We believe that once these improvements are clearly reflected, the significance of the work will be more evident, and we hope that the improved clarity will enable a more accurate assessment of the work’s contributions.

Our goal in the rebuttal is not only to address each comment, but to do so in a way that meaningfully strengthens the work. We are fully committed to providing careful and constructive responses to every point raised by the reviewers. We will soon provide an updated manuscript and detailed, reviewer-specific replies, and we sincerely hope that the effort we are putting into improving the clarity and presentation of the work will be taken into consideration during re-evaluation. We are truly grateful for the reviewers’ time, attention, and thoughtful engagement.

---

### Author Response · Authors · 2025-11-29

We express our sincere gratitude to **Reviewers BYpE, qaEw, CrB3, and 9qjG** for their insightful and constructive feedback. We are genuinely encouraged by the consensus on the core contributions of our work:

* **Novelty & Innovation:** Reviewers **qaEw** and **CrB3** highlighted the framework as *innovative*, *novel*, and *insightful*, appreciating the original theoretical perspective on model self-evaluation.
* **Theoretical Significance:** Reviewer **BYpE** recognized the value of connecting discriminative self-assessment with *provable generalization bounds*, while Reviewer **qaEw** praised the study as *ambitious and timely* for the broader agenda of trustworthy AI.
* **Empirical Breadth:** Reviewers **BYpE** and **CrB3** acknowledged the *extensive experimental setup* and the systematic evaluation across eleven diverse open-source models.

We fully acknowledge that the initial manuscript required deeper theoretical formalization and clearer exposition to fully substantiate these contributions. **Driven by collective feedback, we have executed a comprehensive revision of the manuscript.** The key improvements are summarized below:

* **Elevated Mathematical Rigor:** Addressing the concerns of **Reviewers CrB3 and qaEw**, we have significantly strengthened the theoretical foundations. We formally defined the hypothesis spaces (**Section 3.2**), provided complete derivations for generalization bounds (**Appendices C, D, and E**), and standardized all mathematical notation via a new **Notation Table (Table 2)** to resolve ambiguities regarding probability terms and constraints.

* **Reinforced Empirical Robustness:** To address **Reviewer BYpE’s** request for standard benchmarks and **Reviewer qaEw’s** comments on objectivity, we have explicitly highlighted our evaluation on the **MATH-500 benchmark** (demonstrating consistent trade-offs) and added a rigorous **Judge Correlation Study (Appendix M)**. These additions confirm that our findings are structural properties of current LLMs, independent of specific datasets or judge bias.

* **Clarified Contributions & Scope:** In response to **Reviewer 9qjG**, we have sharpened the distinction between our work (metacognitive measurement) and performance optimization methods (e.g., ASC, RPC), and explicitly framed self-evaluation as the **"atomic unit of autonomy" (Section 5.4)**. This clarifies the practical utility of C-PVC as a building block for autonomous systems.

We expect these revisions have transformed the paper into a solid theoretical contribution that effectively bridges the gap between statistical learning theory and autonomous reasoning reliability. We hope that the updated manuscript and our detailed individual responses satisfactorily address the concerns raised.

---

### Meta-Review · Area_Chair_D9Ln · 2026-01-08

**Summary:**

This paper proposes a framework that enables LLMs to self-assess reliability. Among the four reviewers, only one gave a positive score, while the other three provided negative scores. All four reviewers found the writing extremely unclear, noting that many methodological and theoretical details are insufficiently explained, and that parts of the theory rely more on intuition than on rigorous justification. In addition, the experimental results were not found to be convincing.

**Reviewer Concerns:**

During the rebuttal, the authors provided further clarification of their experimental results and theoretical claims. However, the motivation for the method and key experimental details remain vague, and the overall presentation is still not sufficiently clear.

**Reviewer Scores:**

The rebuttal might lead Reviewer BYpE to increase their score, but the paper’s overall ratings and quality still appear insufficient for acceptance.

---

### Decision · Program_Chairs · 2026-01-26

Reject